# TopP&R: Robust Support Estimation Approach for Evaluating Fidelity and Diversity in Generative Models

**Pum Jun Kim**[1]    **Yoojin Jang**[1]    **Jisu Kim**[2,3,4]    **Jaejun Yoo**[1]
[1]Ulsan National Institute of Science and Technology
[2]Seoul National University    [3]Inria    [4]Paris-Saclay University
{pumjun.kim,softjin,jaejun.yoo}@unist.ac.kr
jkim82133@snu.ac.kr

## Abstract

We propose a robust and reliable evaluation metric for generative models called Topological Precision and Recall (`TopP&R`, pronounced "topper"), which systematically estimates supports by retaining only topologically and statistically significant features with a certain level of confidence. Existing metrics, such as Inception Score (IS), Fréchet Inception Distance (FID), and various `Precision` and `Recall` (`P&R`) variants, rely heavily on support estimates derived from sample features. However, the reliability of these estimates has been overlooked, even though the quality of the evaluation hinges entirely on their accuracy. In this paper, we demonstrate that current methods not only fail to accurately assess sample quality when support estimation is unreliable, but also yield inconsistent results. In contrast, `TopP&R` reliably evaluates the sample quality and ensures statistical consistency in its results. Our theoretical and experimental findings reveal that `TopP&R` provides a robust evaluation, accurately capturing the true trend of change in samples, even in the presence of outliers and non-independent and identically distributed (Non-IID) perturbations where other methods result in inaccurate support estimations. To our knowledge, `TopP&R` is the first evaluation metric specifically focused on the robust estimation of supports, offering statistical consistency under noise conditions.

## 1   Introduction

In keeping with the remarkable improvements of deep generative models [1, 2, 3, 4, 5, 6, 7, 8, 9], evaluation metrics that can well measure the performance of generative models have also been continuously developed [10, 11, 12, 13, 14]. For instance, Inception Score (IS) [10] measures the Kullback-Leibler divergence between the real and fake sample distributions. Fréchet Inception Score (FID) [11] calculates the distance between the real and fake supports using the estimated mean and variance under the multi-Gaussian assumption. The original Precision and Recall [12] and its variants [13, 14] measure fidelity and diversity separately, where fidelity is about how closely the generated samples resemble real samples, while diversity is about whether a generative model can generate samples that are as diverse as real samples.

Considering the eminent progress of deep generative models based on these existing metrics, some may question why we need another evaluation study. In this paper, we argue that we need more reliable evaluation metrics now precisely, because deep generative models have reached sufficient maturity and evaluation metrics are saturated (Table 8 in [15]). Even more, it has been recently reported that even the most widely used evaluation metric, FID, sometimes doesn't match with the expected perceptual quality, fidelity, and diversity, which means the metrics are not always working properly [16].

37th Conference on Neural Information Processing Systems (NeurIPS 2023).

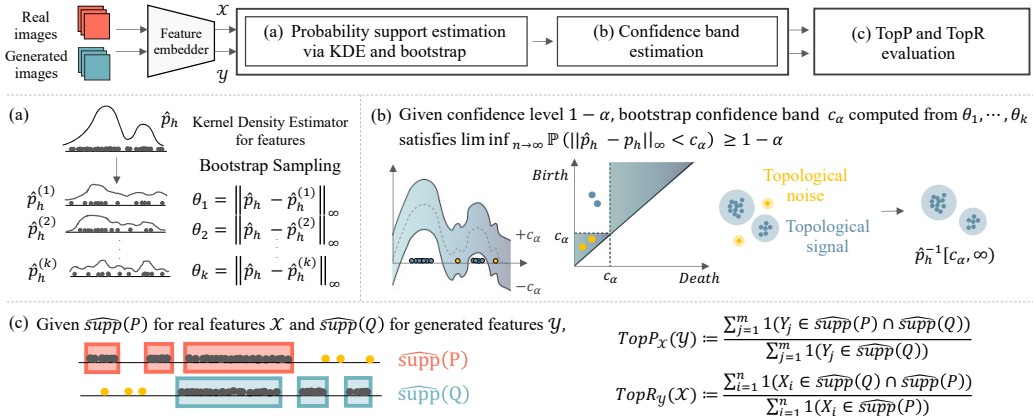

Figure 1: **Illustration of the proposed evaluation pipeline.** (a) Confidence band estimation in Section 2, (b) Robust support estimation, and (c) Evaluation via TopP&R in Section 3.

In addition, existing metrics are vulnerable to the existence of noise because all of them rely on the assumption that real data is clean. However, in practice, real data often contain lots of artifacts, such as mislabeled data and adversarial examples [17, 18], which can cause the overestimation of the data distribution in the evaluation pipeline. This error seriously perturbs the scores, leading to a false impression of improvement when developing generative models. See Appendix G.2 for possible scenarios. Thus, to provide more comprehensive ideas for improvements and to illuminate a new direction in the generative field, we need a more robust and reliable evaluation metric.

An ideal evaluation metric should effectively capture significant patterns (signal) within the data, while being robust against insignificant or accidental patterns (noise) that may arise from factors such as imperfect embedding functions, mislabeled data in the real samples, and other sources of error. Note that there is an inherent tension in developing metrics that meet these goals. On one hand, the metric should be sensitive enough so that it can capture real signals lurking in data. On the other hand, it must ignore noises that hide the signal. However, sensitive metrics are inevitably susceptible to noise to some extent. To address this, one needs a systematic way to answer the following three questions: 1) What is signal and what is noise? 2) How do we draw a line between them? 3) How confident are we on the result?

One solution can be to use the idea of Topological Data Analysis (TDA) [19] and statistical inference. TDA is a recent and emerging field of data science that relies on topological tools to infer relevant features for possibly complex data. A key object in TDA is persistent homology, which observes how long each topological feature would survive over varying resolutions and provides a measure to quantify its significance; *i.e.*, if some features persist longer than others over varying resolutions, we consider them as topological signals and vice versa as noise.

In this paper, we combine these ideas to estimate supports more robustly and overcome various issues of previous metrics such as unboundedness, inconsistency, etc. Our main contributions are as follows: we establish 1) a systematic approach to estimate supports via Kernel Density Estimator (KDE) derived under topological conditions; 2) a new metric that is robust to outliers while reliably detecting the change of distributions on various scenarios; 3) a consistency guarantee with robustness under very weak assumptions that are suitable for high dimensional data; 4) a combination of a noise framework and a statistical inference in TDA. Our code is available at TopP&R-NeurIPS 2023.

## 2 Background

To lay the foundation for our method and theoretical analysis, we first explain the previous evaluation method Precision and Recall (P&R). Then, we introduce the main idea of persistent homology and its confidence estimation techniques that bring the benefit of using topological and statistical tools for addressing uncertainty in samples. For the sake of space constraints and to streamline the discussion of our main idea, we only provide a brief overview of the concepts that are relevant to this work and refer the reader to Appendix A or [20, 21, 22, 23] for further details on TDA.

**Notation.** For any $x$ and $r > 0$, we use the notation $\mathcal{B}_d(x, r) = \{y : d(y, x) < r\}$ be the open ball in distance $d$ of radius $r$. We also write $\mathcal{B}(x, r)$ when $d$ is understood from the context. For a distribution $P$ on $\mathbb{R}^d$, we let $\text{supp}(P) := \{x \in \mathbb{R}^d : P(\mathcal{B}(x, r)) > 0 \text{ for all } r > 0\}$ be the support of $P$. Throughout the paper, we refer to $\text{supp}(P)$ as support of $P$, or simply support, or manifold, but we don't necessarily require the (geometrical) manifold structure on $\text{supp}(P)$. Note that, when the support is high dimensional, what we can recover at most through estimation is the partial support. For a kernel function $K : \mathbb{R}^d \to \mathbb{R}$, a dataset $\mathcal{X} = \{X_1, \ldots, X_n\} \subset \mathbb{R}^d$ and bandwidth $h > 0$, we let the kernel density estimator (KDE) as $\hat{p}_h(x) := \frac{1}{nh^d} \sum_{i=1}^{n} K\left(\frac{x - X_i}{h}\right)$, and we let the average KDE as $p_h := \mathbb{E}[\hat{p}_h]$. We denote by $P$, $Q$ the probability distributions in $\mathbb{R}^d$ of real data and generated samples, respectively. And we use $\mathcal{X} = \{X_1, \ldots, X_n\} \subset \mathbb{R}^d$ and $\mathcal{Y} = \{Y_1, \ldots, Y_m\} \subset \mathbb{R}^d$ for real data and generated samples possibly with noise, respectively.

**Precision and Recall.** There exist two aspects of generative samples' quality; fidelity and diversity. Fidelity refers to the degree to which the generated samples resemble the real ones. Diversity, on the other hand, measures whether the generated samples cover the full variability of the real samples. Sajjadi et al. [12] was the first to propose assessing these two aspects separately via `Precision` and `Recall` (P&R). In the ideal case where we have full access to the probability distributions $P$ and $Q$, $\text{precision}_P(Q) := Q(\text{supp}(P))$, $\text{recall}_Q(P) := P(\text{supp}(Q))$, which correspond to the max `Precision` and max `Recall` in [12], respectively.

**Persistent homology and diagram.** Persistent homology is a tool in computational topology that measures the topological invariants (homological features) of data that persist across multiple scales, and is represented in the persistence diagram. Formally, let *filtration* be a collection of subspaces $\mathcal{F} = \{\mathcal{F}_\delta \subset \mathbb{R}^d\}_{\delta \in \mathbb{R}}$ with $\delta_1 \leq \delta_2$ implying $\mathcal{F}_{\delta_1} \subset \mathcal{F}_{\delta_2}$. Typically, filtration is defined through a function $f$ related to data. Given a function $f : \mathbb{R}^d \to \mathbb{R}$, we consider its sublevel filtration $\{f^{-1}(-\infty, \delta]\}_{\delta \in \mathbb{R}}$ or superlevel filtration $\{f^{-1}[\delta, \infty)\}_{\delta \in \mathbb{R}}$. For a filtration $\mathcal{F}$ and for each nonnegative $k$, we track when $k$-dimensional homological features (*e.g.*, 0-dimension: connected component, 1-dimension: loop, 2-dimension: cavity, ...) appear and disappear. As $\delta$ increases or decreases in the filtration $\{\mathcal{F}_\delta\}$, if a homological feature appears at $\mathcal{F}_b$ and disappears at $\mathcal{F}_d$, we say that it is born at $b$ and dies at $d$. By considering these pairs $\{(b, d)\}$ as points in the plane $(\mathbb{R} \cup \{\pm\infty\})^2$, we obtain a *persistence diagram*. From this, a homological feature with a longer life length, $d - b$, can be treated as a significant feature, and a homological feature with a shorter life length as a topological noise, which lies near the diagonal line $\{(\delta, \delta) : \delta \in \mathbb{R}\}$ (Figure 1 (b)).

**Confidence band estimation.** Statistical inference has recently been developed for TDA [24, 25, 26]. TDA consists of features reflecting topological characteristics of data, and it is of question to systematically distinguish features that are indeed from geometrical structures and features that are insignificant or due to noise. To *statistically* separate topologically significant features from topological noise, we employ confidence band estimation. Given the significance level $\alpha$, let confidence band $c_\alpha$ be the bootstrap bandwidth of $\|\hat{p}_h - p_h\|_\infty$, computed as in Algorithm 1 (see Appendix H.2). Then it satisfies $\liminf_{n \to \infty} \mathbb{P}(\|\hat{p}_h - p_h\|_\infty < c_\alpha) \geq 1 - \alpha$, as in Proposition D.2 (see Appendix D). This confidence band can simultaneously determine significant topological features while filtering out noise features. We use $c_\mathcal{X}$ and $c_\mathcal{Y}$ to denote the confidence band defined under significance level $\alpha$ according to the datasets $\mathcal{X}$ and $\mathcal{Y}$. In later sections, we use these tools to provide a more rigorous way of scoring samples based on the confidence level we set.

## 3 Robust support estimation for reliable evaluation

Current evaluation metrics for generative models typically rely on strong regularity conditions. For example, they assume samples are well-curated without outliers or adversarial perturbation, real or generative models have bounded densities, etc. However, practical scenarios are wild: both real and generated samples can be corrupted with noise from various sources, and the real data can be very sparsely distributed without density. In this work, we consider more general and practical situations, wherein both real and generated samples can have noises coming from the sampling procedure, remained uncertainty due to data or model, etc. See Appendix G.2 for more on practical scenarios.

**Overview of our metric** We design our metric to evaluate the performance very conservatively. Our metric is based on topologically significant data structures with statistical confidence above a certain level. Toward this, we apply KDE as a function $f$ to define a filtration, which allows

us to approximate the support with data through $\{f^{-1}[\delta, \infty)\}_{\delta \in \mathbb{R}}$. Since the significance of data comprising the support is determined by the life length of homological features, we calculate $c_\alpha$ that enables us to systematically separate short/long lifetimes of homological features. We then estimate the supports with topologically significant data structure via superlevel set $f^{-1}[c_\alpha, \infty)$ and finally, we evaluate fidelity and diversity with the estimated supports. We have collectively named this process TopP&R. By its nature, TopP&R is bounded and yields consistent performance under various conditions such as noisy data points, outliers, and even with long-tailed data distribution.

## 3.1 Topological precision and recall

To facilitate our discussion, we rewrite the precision in Section 2 as $\text{precision}_P(\mathcal{Y}) = Q(\text{supp}(P) \cap \text{supp}(Q))/Q(\text{supp}(Q))$ and define the precision of data points as

$$\text{precision}_P(\mathcal{Y}) := \frac{\sum_{j=1}^m 1\left(Y_j \in \text{supp}(P) \cap \text{supp}(Q)\right)}{\sum_{j=1}^m 1\left(Y_j \in \text{supp}(Q)\right)}, \tag{1}$$

which is just replacing the distribution $Q$ with the empirical distribution $\frac{1}{m}\sum_{j=1}^m \delta_{Y_j}$ of $Y$ in the precision. Similarly,

$$\text{recall}_Q(\mathcal{X}) := \frac{\sum_{i=1}^n 1\left(X_i \in \text{supp}(Q) \cap \text{supp}(P)\right)}{\sum_{i=1}^n 1\left(X_i \in \text{supp}(P)\right)}. \tag{2}$$

In practice, $\text{supp}(P)$ and $\text{supp}(Q)$ are not known a priori and need to be estimated, and these estimates should be robust to noise since we allow it now. For this, we use the KDE $\hat{p}_{h_n}(x) := \frac{1}{nh_n^d}\sum_{i=1}^n K\left(\frac{x-X_i}{h_n}\right)$ of $\mathcal{X}$ and the bootstrap bandwidth $c_\mathcal{X}$ of $\|\hat{p}_{h_n} - p_{h_n}\|_\infty$, where $h_n > 0$ and a significance level $\alpha \in (0,1)$ (Section 2). Then, we estimate the support of $P$ by the superlevel set at $c_\mathcal{X}$ [1] as $\hat{\text{supp}}(P) = \hat{p}_{h_n}^{-1}[c_\mathcal{X}, \infty)$, which allows to filter out noise whose KDE values are likely to be small. Similarly, the support of $Q$ is estimated: $\hat{\text{supp}}(Q) = \hat{q}_{h_m}^{-1}[c_\mathcal{Y}, \infty)$, where $\hat{q}_{h_m}(x) := \frac{1}{mh_m^d}\sum_{j=1}^m K\left(\frac{x-Y_j}{h_m}\right)$ is the KDE of $\mathcal{Y}$ and $c_\mathcal{Y}$ is the bootstrap bandwidth of $\|\hat{q}_{h_m} - q_{h_m}\|_\infty$

For the robust estimates of the precision, we apply the support estimates to $\text{precision}_P(\mathcal{Y})$ and $\text{recall}_Q(\mathcal{X})$ and define the topological precision and recall (TopP&R) as

$$\text{TopP}_\mathcal{X}(\mathcal{Y}) := \frac{\sum_{j=1}^m 1\left(Y_j \in \hat{\text{supp}}(P) \cap \hat{\text{supp}}(Q)\right)}{\sum_{j=1}^m 1\left(Y_j \in \hat{\text{supp}}(Q)\right)}$$

$$= \frac{\sum_{j=1}^m 1\left(\hat{p}_{h_n}(Y_j) > c_\mathcal{X}, \ \hat{q}_{h_m}(Y_j) > c_\mathcal{Y}\right)}{\sum_{j=1}^m 1\left(\hat{q}_{h_m}(Y_j) > c_\mathcal{Y}\right)}, \tag{3}$$

$$\text{TopR}_\mathcal{Y}(\mathcal{X}) := \frac{\sum_{i=1}^n 1\left(\hat{q}_{h_m}(X_i) > c_\mathcal{Y}, \ \hat{p}_{h_n}(X_i) > c_\mathcal{X}\right)}{\sum_{i=1}^n 1\left(\hat{p}_{h_n}(X_i) > c_\mathcal{X}\right)}. \tag{4}$$

The kernel bandwidths $h_n$ and $h_m$ are hyperparameters, and we provide guidelines to select the optimal bandwidths $h_n$ and $h_m$ in practice (See Appendix H.4).

## 3.2 Bandwidth estimation using bootstrapping

Using the bootstrap bandwidth $c_\mathcal{X}$ as the threshold is the key part of our estimator (TopP&R) for robustly estimating $\text{supp}(P)$. As we have seen in Section 2, the bootstrap bandwidth $c_\mathcal{X}$ filters out the topological noise in topological data analysis. Analogously, using $c_\mathcal{X}$ allows to robustly estimate $\text{supp}(P)$. When $X_i$ is an outlier, its KDE value $\hat{p}_h(X_i)$ is likely to be small as well as the values at the connected component generated by $X_i$. So those components from outliers are likely to be removed in the estimated support $\hat{p}_h^{-1}[c_\mathcal{X}, \infty)$. Higher dimensional homological noises are also removed. Hence, the estimated support denoises topological noise and robustly estimates $\text{supp}(P)$. See Appendix C for a more detailed explanation.

Now that we are only left with topological features of high confidence, this allows us to draw analogies to confidence intervals in statistical analysis, where the uncertainty of the samples is treated

---

[1]The computation of $c_\alpha$ and its practical interpretation is described in Algorithm 1.

by setting the level of confidence. In the next section, we show that TopP&R not only gives a more reliable evaluation score for generated samples but also has good theoretical properties.

### 3.3 Addressing the curse of dimensionality

As discussed in Section 3.2, getting the bootstrap bandwidth $c_{\mathcal{X}}$ with a theoretical guarantee plays a key role in our metric, and the choice of KDE as a filtration function is inevitable, as in Remark 4.4. However, computing the support of a high-dimensional feature with KDE demands significant computation, and the accuracy is low due to low density values. This hinders an efficient and correct evaluation in practice. To address this issue, we apply a random projection into a low-dimensional space by leveraging the Johnson-Lindenstrauss Lemma (Lemma B.1). This lemma posits that using a random projection effectively preserves information regarding distances and homological features, composed of high-dimensional features, in a low-dimensional representation. Furthermore, we have shown that random projection does not substantially reduce the influence of noise, nor does it affect the performance of TopP&R under various conditions with complex data (Section 5, I.7, and I.8).

## 4  Consistency with robustness of TopP&R

The key property of TopP&R is consistency with robustness. The consistency ensures that, the precision and the recall we compute from the *data* approaches the precision and the recall from the *distribution* as we have more samples. The consistency allows to investigate the precision and recall of full distributions only with access to finite sampled data. TopP&R achieves consistency with robustness, that is, the consistency holds with the data possibly corrupted by noise. This is due to the robust estimation of supports with KDE with confidence bands.

We demonstrate the statistical model for both data and noise. Let $P, Q, \mathcal{X}, \mathcal{Y}$ be as in Section 2, and let $\mathcal{X}^0, \mathcal{Y}^0$ be real data and generated data without noise. $\mathcal{X}, \mathcal{Y}, \mathcal{X}^0, \mathcal{Y}^0$ are understood as multisets, *i.e.*, elements can be repeated. We first assume that the uncorrupted data are IID.

**Assumption 1.** *The data $\mathcal{X}^0 = \{X_1^0, \ldots, X_n^0\}$ and $\mathcal{Y}^0 = \{Y_1^0, \ldots, Y_m^0\}$ are IID from $P$ and $Q$, respectively.*

In practice, the data is often corrupted with noise. We consider the adversarial noise, where some fraction of data are replaced with arbitrary point cloud data.

**Assumption 2.** *Let $\{\rho_k\}_{k \in \mathbb{N}}$ be a sequence of nonnegative real numbers. Then the observed data $\mathcal{X}$ and $\mathcal{Y}$ satisfies $\left|\mathcal{X} \backslash \mathcal{X}^0\right| = n\rho_n$ and $\left|\mathcal{Y} \backslash \mathcal{Y}^0\right| = m\rho_m$.*

In the adversarial model, we control the level of noise by the fraction $\rho$, but do not assume other conditions such as IID or boundedness, to make our noise model very general and challenging.

For distributions and kernel functions, we assume weak conditions, detailed in Assumption A1 and A2 in Appendix D. Under the data and the noise models, TopP&R achieves consistency with robustness. That is, the estimated precision and recall are asymptotically correct with high probability even if up to a portion of $1/\sqrt{n}$ or $1/\sqrt{m}$ are replaced by adversarial noise. This is due to the robust estimation of the support with the kernel density estimator with the confidence band of the persistent homology.

**Proposition 4.1.** *Suppose Assumption 1, 2, A1, A2 hold. Suppose $\alpha \to 0$, $h_n \to 0$, $nh_n^d \to \infty$, $nh_n^{-d}\rho_n^2 \to 0$, and similar relations hold for $h_m$, $\rho_m$. Then,*
$$\left|\mathrm{TopP}_{\mathcal{X}}(\mathcal{Y}) - \mathrm{precision}_P(\mathcal{Y})\right| = O_{\mathbb{P}}\left(Q(B_{n,m}) + \rho_m\right),$$
$$\left|\mathrm{TopR}_{\mathcal{Y}}(\mathcal{X}) - \mathrm{recall}_Q(\mathcal{X})\right| = O_{\mathbb{P}}\left(P(A_{n,m}) + \rho_n\right),$$
*for fixed sequences of sets $\{A_{n,m}\}_{n,m \in \mathbb{N}}, \{B_{n,m}\}_{n,m \in \mathbb{N}}$ with $P(A_{n,m}) \to 0$ and $Q(B_{n,m}) \to 0$ as $n, m \to \infty$.*

**Theorem 4.2.** *Under the same condition as in Proposition 4.1,*
$$\left|\mathrm{TopP}_{\mathcal{X}}(\mathcal{Y}) - \mathrm{precision}_P(Q)\right| = O_{\mathbb{P}}\left(Q(B_{n,m}) + \rho_m\right),$$
$$\left|\mathrm{TopR}_{\mathcal{Y}}(\mathcal{X}) - \mathrm{recall}_Q(P)\right| = O_{\mathbb{P}}\left(P(A_{n,m}) + \rho_n\right).$$

Since $P(A_{n,m}) \to 0$ and $Q(B_{n,m}) \to 0$, these imply consistencies of TopP&R. In fact, additionally under minor probabilistic and geometrical assumptions, $P(A_{n,m})$ and $Q(B_{n,m})$ are of order $h_m + h_n$.

**Lemma 4.3.** *Under the same condition as in Proposition 4.1 and additionally under Assumption A3, A4, $P(A_{n,m}) = O(h_n + h_m)$ and $Q(B_{n,m}) = O(h_n + h_m)$.*

*Remark* 4.4. Consistency guarantees from Proposition 4.1 and Theorem 4.2 are in principle due to the uniform convergence of KDE over varying bandwidth $h_n$ (Proposition D.2). Once we replace estimating the support with KDE by k-NN or something else, we wouldn't have consistency guarantees. Hence, using the KDE is an essential part for the theoretical guarantees of `TopP&R`.

Our theoretical results in Proposition 4.1 and Theorem 4.2 are novel and important in several perspectives. These results are among the first theoretical guarantees for evaluation metrics for generative models as far as we are aware of. Also, as in Remark D.1, assumptions are very weak and suitable for high dimensional data. Also, robustness to adversarial noise is provably guaranteed.

## 5 Experiments

A good evaluation metric should not only possess desirable theoretical properties but also effectively capture the changes in the underlying data distribution. To examine the performance of evaluation metrics, we carefully select a set of experiments for sanity checks. With toy and real image data, we check 1) how well the metric captures the true trend of underlying data distributions and 2) how well the metric resists perturbations applied to samples.

We compare `TopP&R` with several representative evaluation metrics that include Improved `Precision` and `Recall` (P&R) [13], `Density` and `Coverage` (D&C) [14], Geometric Component Analysis (GCA) [27], and Manifold Topology Divergence (MTD) [28] (Appendix F). Both GCA and MTD are the recent evaluation metrics that utilize topological features to some extent; GCA defines `precision` and `recall` based on connected components of $P$ and $Q$, and MTD measures the distance between two distributions using the sum of lifetimes of homological features. For all the experiments, linear random projection to 32 dimensions is additionally used for `TopP&R`, and the shaded area of all figures denotes the $\pm 1$ standard deviation for ten trials. For a fair comparison with existing metrics, we have utilized 10k real and fake samples for all experiments. For more details, please refer to Appendix H.1.

### 5.1 Sanity checks with toy data

Following [14], we first examine how well the metric reflects the trend of $\mathcal{Y}$ moving away from $\mathcal{X}$ and whether it is suitable for finding mode-drop phenomena. In addition to these, we newly design several experiments that can highlight `TopP&R`'s favorable theoretical properties of consistency with robustness in various scenarios.

#### 5.1.1 Shifting the generated feature manifold

We generate samples from $\mathcal{X} \sim \mathcal{N}(\mathbf{0}, I)$ and $\mathcal{Y} \sim \mathcal{N}(\mu\mathbf{1}, I)$ in $\mathbb{R}^{64}$, where $\mathbf{1}$ is a vector of ones and $I$ is an identity matrix. We examine how each metric responds to shifting $\mathcal{Y}$ with $\mu \in [-1, 1]$ while there are outliers at $\mathbf{3} \in \mathbb{R}^{64}$ for both $\mathcal{X}$ and $\mathcal{Y}$ (Figure 2). We discovered that GCA struggles to detect changes using its default hyperparameter configuration, and this issue persists even after performing an exhaustive hyperparameter sweep. Since empirical tuning of the hyperparameters is required for each dataset, utilizing GCA in practical applications proves to be challenging (Appendix F). Both improved P&R and D&C behave pathologically since these methods estimate the support via the k-nearest neighbor algorithm, which inevitably overestimate the underlying support when there are outliers. For example, when $\mu < 0.5$, `Recall` returns a high-diversity score, even though the true supports of $\mathcal{X}$ and $\mathcal{Y}$ are actually far apart. In addition, P&R does not reach 1 in high dimensions even when $\mathcal{X} = \mathcal{Y}$. D&C [14] yields better results than P&R because it consistently uses $\mathcal{X}$ (the real data distribution) as a reference point, which typically has fewer outliers than $\mathcal{Y}$ (the fake data distribution). However, there is no guarantee that this will always be the case in practice [17, 18]. If an outlier is present in $\mathcal{X}$, D&C also returns an incorrect high-fidelity score at $\mu > 0.5$. On the other hand, `TopP&R` shows a stable trend unaffected by the outlier, demonstrating its robustness.

#### 5.1.2 Dropping modes

We simulate mode-drop phenomena by gradually dropping all but one mode from the fake distribution $\mathcal{Y}$ that is initially identical to $\mathcal{X}$ (Figure 3). Here, we consider the mixture of Gaussians with seven

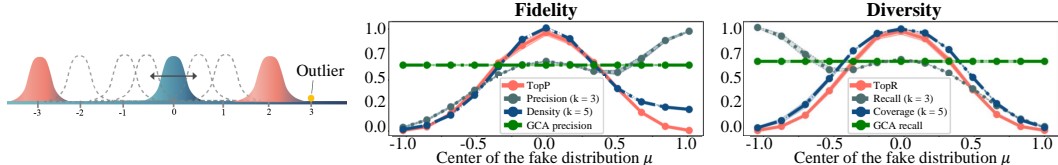

Figure 2: Behaviors of evaluation metrics for outliers on real and fake distribution. For both real and fake data, the outliers are fixed at $3 \in \mathbb{R}^{64}$, and the parameter $\mu$ is shifted from -1 to 1.

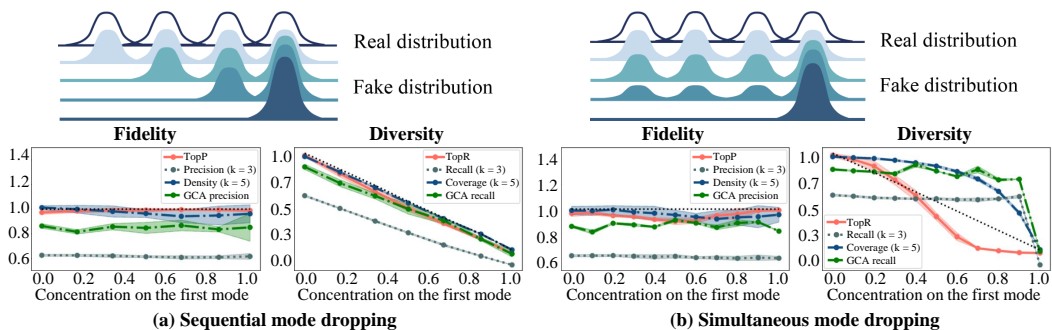

Figure 3: Behaviors of evaluation metrics for (a) sequential and (b) simultaneous mode-drop scenarios. The horizontal axis shows the concentration ratio on the distribution centered at $\mu = 0$.

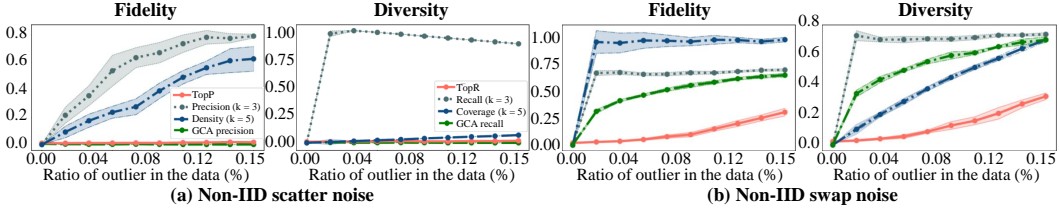

Figure 4: Behaviors of evaluation metrics on Non-IID perturbations. We replace a certain percentage of real and fake data (a) with random uniform noise or (b) by switching some of real and fake data.

modes in $\mathbb{R}^{64}$. We keep the number of samples in $\mathcal{X}$ constant so that the same amount of decreased samples are supplemented to the first mode which leads fidelity to be fixed to 1. We observe that the values of `Precision` fail to saturate, *i.e.*, mainly smaller than 1, and the `Density` fluctuates to a value greater than 1, showing their instability and unboundedness. `Recall` and GCA do not respond to the simultaneous mode drop, and `Coverage` decays slowly compared to the reference line. In contrast, `TopP` performs well, being held at the upper bound of 1 in sequential mode drop, and `TopR` also decreases closest to the reference line in simultaneous mode drop.

### 5.1.3 Tolerance to Non-IID perturbations

Robustness to perturbations is another important aspect we should consider when designing a metric. Here, we test whether `TopP&R` behaves stably under two variants of noise cases (see Section G.3); 1) **scatter noise**: replacing $X_i$ and $Y_j$ with uniformly distributed noise and 2) **swap noise**: swapping the position between $X_i$ and $Y_j$. These two cases all correspond to the adversarial noise model of Assumption 2. We set $\mathcal{X} \sim \mathcal{N}(\mu = 0, I) \in \mathbb{R}^{64}$ and $\mathcal{Y} \sim \mathcal{N}(\mu = 1, I) \in \mathbb{R}^{64}$ where $\mu = 1$, and thus an ideal evaluation metric must return zero for both fidelity and diversity. In the result, while the GCA precision is relatively robust to the scatter noise, GCA recall tends to be sensitive to the swap noise. In both cases, we find that `P&R` and `D&C` are more sensitive while `TopP&R` remains relatively stable until the noise ratio reaches $15\%$ of the total data, which is a clear example of the weakness of existing metrics to perturbation (Figure 4).

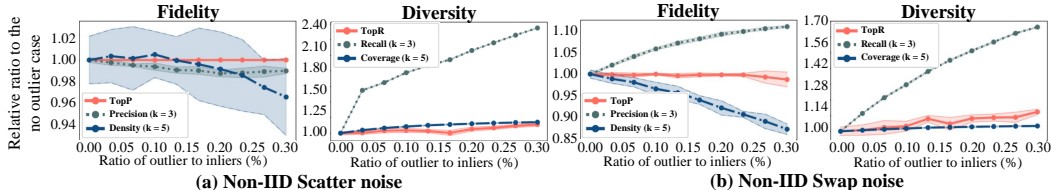

**(a) Non-IID Scatter noise**  **(b) Non-IID Swap noise**

Figure 5: Comparison of evaluation metrics on Non-IID perturbations using FFHQ dataset. We replaced certain ratio of $\mathcal{X}$ and $\mathcal{Y}$ (a) with outliers and (b) by switching some of real and fake features.

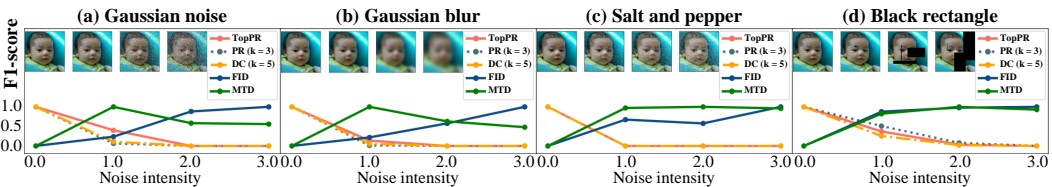

Figure 6: Verification of whether `TopP&R` can make an accurate quantitative assessment of noisy image features. Gaussian noise, gaussian blur, salt and pepper, and black rectangle noise are added to the FFHQ images and embedded with T4096.

## 5.2 Sanity check with Real data

Now that we have verified the metrics on toy data, we move on to real data. Just like in the toy experiments, we concentrate on how the metrics behave in extreme situations, such as outliers, mode-drop phenomena, perceptual distortions, etc. We also test different image embedders, including pretrained VGG16 [29], InceptionV3 [30], and SwAV [31].

### 5.2.1 Dropping modes in Baby ImageNet

We have conducted an additional experiment using Baby ImageNet [15] to investigate the sensitivity of `TopP&R` to mode-drop in real-world data. The performance of each metric (Figure A4) is measured with the identical data while simultaneously dropping the modes of nine classes of Baby ImageNet, in total of ten classes. Since our experiment involves gradually reducing the fixed number of fake samples until nine modes of the fake distribution vanish, the ground truth diversity should decrease linearly. From the experimental results, consistent to the toy result of Figure 3, both `D&C` and `P&R` still struggle to respond to simultaneous mode dropping. In contrast, `TopP&R` consistently exhibit a high level of sensitivity to subtle distribution changes. This notable capability of `TopP&R` can be attributed to its direct approximation of the underlying distribution, distinguishing it from other metrics. In addition, we perform the experiments on a dataset with long-tailed distribution and find that `TopP&R` captures the trend well even when there are minority sets (Appendix I.1). This again shows the reliability of `TopP&R`.

### 5.2.2 Robustness to perturbations

To test the robustness of our metric against the adversarial noise model of Assumption 2, we test both scatter-noise and swap noise scenarios with real data (see Section G.3). In the experiment, following Kynkäänniemi et al. [13], we first classify inliers and outliers that are generated by StyleGAN [1]. For scatter noise we add the outliers to the inliers and for swap noise we swap the real FFHQ images with generated images. Under these specific noise conditions, `Precision` shows similar or even better robustness than `Density` (Figure 5). On the other hand, `Coverage` is more robust than `Recall`. In both cases, `TopP&R` shows the best performance, resistant to noise.

### 5.2.3 Sensitiveness to the noise intensity

One of the advantages of FID [11] is that it is good at estimating the degrees of distortion applied to the images. Similarly, we check whether the F1-score based on `TopP&R` provides a reasonable evaluation according to different noise levels. As illustrated in Figure 6 and A5, $\mathcal{X}$ and $\mathcal{Y}$ are sets

Table 1: Generative models trained on CIFAR-10 are ranked by FID, KID, MTD, and F1-scores based on `TopP&R`, `D&C` and `P&R`, respectively. The $\mathcal{X}$ and $\mathcal{Y}$ are embedded with InceptionV3, VGG16, and SwAV. The number inside the parenthesis denotes the rank based on each metric.

| | Model | StyleGAN2 | ReACGAN | BigGAN | PDGAN | ACGAN | WGAN-GP |
|---|---|---|---|---|---|---|---|
| **InceptionV3** | **FID** ($\downarrow$) | 3.78 (1) | 3.87 (2) | 4.16 (3) | 31.54 (4) | 33.39 (5) | 107.68 (6) |
| | **KID** ($\downarrow$) | 0.002 (1) | 0.012 (3) | 0.011 (2) | 0.025 (4) | 0.029 (5) | 0.137 (6) |
| | **TopP&R** ($\uparrow$) | 0.9769 (1) | 0.8457 (2) | 0.7751 (3) | 0.7339 (4) | 0.6951 (5) | 0.0163 (6) |
| | D&C ($\uparrow$) | 0.9626 (2) | 0.9409 (3) | 1.1562 (1) | 0.4383 (4) | 0.3883 (5) | 0.1913 (6) |
| | P&R ($\uparrow$) | 0.6232 (1) | 0.3320 (2) | 0.3278 (3) | 0.1801 (4) | 0.0986 (5) | 0.0604 (6) |
| | MTD ($\downarrow$) | 2.3380 (3) | 2.2687 (2) | 1.4473 (1) | 7.0188 (4) | 8.0728 (5) | 11.498 (6) |
| **VGG16** | **TopP&R** ($\uparrow$) | 0.9754 (1) | 0.5727 (3) | 0.7556 (2) | 0.4021 (4) | 0.3463 (5) | 0.0011 (6) |
| | D&C ($\uparrow$) | 0.9831 (3) | 1.0484 (1) | 0.9701 (4) | 0.9872 (2) | 0.8971 (5) | 0.6372 (6) |
| | P&R ($\uparrow$) | 0.6861 (1) | 0.1915 (3) | 0.3526 (2) | 0.0379 (4) | 0.0195 (5) | 0.0001 (6) |
| | MTD ($\downarrow$) | 25.757 (4) | 25.826 (3) | 34.755 (5) | 24.586 (2) | 23.318 (1) | 41.346 (6) |
| **SwAV** | **TopP&R** ($\uparrow$) | 0.9093 (1) | 0.3568 (3) | 0.5578 (2) | 0.1592 (4) | 0.1065 (5) | 0.0003 (6) |
| | D&C ($\uparrow$) | 1.0732 (1) | 0.9492 (3) | 1.0419 (2) | 0.6328 (4) | 0.4565 (5) | 0.0721 (6) |
| | P&R ($\uparrow$) | 0.5623 (1) | 0.0901 (3) | 0.1459 (2) | 0.0025 (4) | 0.0000 (6) | 0.0002 (5) |
| | MTD ($\downarrow$) | 1.1098 (1) | 1.5512 (3) | 1.3280 (2) | 1.8302 (4) | 2.2982 (5) | 4.9378 (6) |

of reference and noisy features, respectively. The experimental results show that `TopP&R` actually reflects well the different degrees of distortion added to the images while a similar topology-based method MTD shows inconsistent behavior to the distortions.

### 5.2.4 Ranking between generative models

The alignment between FID (or KID) and perceptual evaluation has been well-established in prior research, and these scores are widely used as a primary metric in the development of generative models due to its close correspondence with human perception. Consequently, generative models have evolved to align with FID's macroscopic perspective. Therefore, we believed that the order determined by FID at a high level captures to some extent the true performance hierarchy among models, even if it may not perfectly reflect it. In other words, if the development of generative models based on FID leads to genuine improvements in generative performance and if there is a meaningful correlation, similar rankings should be maintained even when the representation or embedding model changes. From this standpoint, while other metrics exhibit fluctuating rankings, `TopP&R` consistently provides the most stable and consistent results similar to both FID and KID. To quantitatively compare the similarity of rankings across varying embedders by different metrics, we have computed mean Hamming Distance (MHD) (Appendix H.6) where lower value indicates more similarity. `TopP&R`, `P&R`, `D&C`, and MTD have MHDs of 1.33, 2.66, 3.0, and 3.33, respectively.

## 6 Conclusions

Many works have been proposed recently to assess the fidelity and diversity of generative models. However, none of them has focused on the accurate estimation of support even though it is one of the key components in the entire evaluation pipeline. In this paper, we proposed topological precision and recall (`TopP&R`) that provides a systematical fix by robustly estimating the support with both topological and statistical treatments. To the best of our knowledge, `TopP&R` is the first evaluation metric that offers statistical consistency under noisy conditions, which may arise in real practice. Our theoretical and experimental results showed that `TopP&R` serves as a robust and reliable evaluation metric under various embeddings and noise conditions, including mode drop, outliers, and Non-IID perturbations. Last but not least, `TopP&R` provides the most consistent ranking among different generative models across different embeddings via calculating its F1-score.

## Acknowledgements

This work was supported by the National Research Foundation of Korea (NRF) grant funded by the Korea government (MSIT) (No.2.220574.01), Institute of Information & communications Technology Planning & Evaluation (IITP) grant funded by the Korea government (MSIT) (No.2020-0-01336, Artificial Intelligence Graduate School Program (UNIST), No.2021-0-02068, Artificial Intelligence Innovation Hub, No.2022-0-00959, (Part 2) Few-Shot Learning of Causal Inference in Vision and Language for Decision Making, No.2022-0-00264, Comprehensive Video Understanding and Generation with Knowledge-based Deep Logic Neural Network).

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

# Appendix

## A  More Background on Topological Data Analysis

Topological data analysis (TDA) [19] is a recent and emerging field of data science that relies on topological tools to infer relevant features for possibly complex data. A key object in TDA is persistent homology, which quantifies salient topological features of data by observing them in multi-resolutions.

### A.1  Persistent Homology

**Persistent homnology.** *Persistent homology* is a multiscale approach to represent the topological features. For a filtration $\mathcal{F}$ and for each $k \in \mathbb{N}_0 = \mathbb{N} \cup \{0\}$, the associated $k$-th persistent homology $PH_k\mathcal{F}$ is a collection of $k$-th dimensional homologies $\{H_k\mathcal{F}_\delta\}_{\delta \in \mathbb{R}}$ equipped with homomorphisms $\{\imath_k^{a,b} : H_k\mathcal{F}_a \to H_k\mathcal{F}_b\}_{a \leq b}$ induced by the inclusion $\mathcal{F}_a \subset \mathcal{F}_b$.

**Persistence diagram.** For the $k$-th persistent homology $PH_k\mathcal{F}$, the set of filtration levels at which a specific homology appears is always an interval $[b, d) \subset [-\infty, \infty]$. The corresponding $k$-th persistence diagram is a multiset of points $(\mathbb{R} \cup \{\infty\})^2$, consisting of all pairs $(b, d)$ where $[b, d)$ is the interval of filtration values for which a specific homology appears in $PH_k\mathcal{F}$.

### A.2  Statistical Inference of Persistent Homology

As discussed above, a homological feature with a long life-length is important information in topology while the homology with a short life-length can be treated as non-significant information or noise. The confidence band estimator provides the confidence set from the features that only include topologically and statistically significant (statistically considered as elements in the population set) under a certain level of confidence. And to build a confidence set, we first need to endow a metric on the space of persistence diagrams.

**Bottleneck distance.** The most fundamental metric to measure the distance between two persistence diagrams is the *bottleneck distance*.

**Definition A.1.** The *bottleneck* distance between two persistence diagrams $D_1$ and $D_2$ is defined by

$$d_B(D_1, D_2) = \inf_{\gamma \in \Gamma} \sup_{p \in D_1} \|p - \gamma(p)\|_\infty,$$

where the set $\Gamma$ consists of all the bijections $\gamma : D_1 \cup Diag \to D_2 \cup Diag$, and $Diag$ is the diagonal $\{(x, x) : x \in \mathbb{R}\} \subset \mathbb{R}^2$ with infinite multiplicity.

One way of constructing the confidence set uses the superlevel filtration of the kernel density estimator and the bootstrap confidence band. Let $\mathcal{X} = \{X_1, X_2, ..., X_n\}$ as given points cloud, then the probability for the distribution of points can be estimated via KDE defined as following: $\hat{p}_h(x) := \frac{1}{nh^d} \sum_{i=1}^n K\left(\frac{x-X_i}{h}\right)$ where $h$ is the bandwidth and $d$ as a dimension of the space. We compute $\hat{p}_h$ and $\hat{p}_h^*$, which are the KDE of $\mathcal{X}$ and the KDE of bootstrapped samples $\mathcal{X}^*$, respectively. Now, given the significance level $\alpha$ and $h > 0$, let confidence band $q_\mathcal{X}$ be bootstrap bandwidth of a Gaussian Empirical Process [32, 33], $\sqrt{n}\|\hat{p}_h - \hat{p}_h^*\|_\infty$. Then it satisfies $P(\sqrt{n}\|\hat{p}_h - p_h\|_\infty < q_\mathcal{X}) \geq 1 - \alpha$, as in Proposition D.2 in Section D. Then $\mathcal{B}_{d_B}(\hat{\mathcal{P}}_h, c_\mathcal{X})$, the ball of persistent homology centered at $\hat{\mathcal{P}}_h$ and radius $c_\mathcal{X} = q_\mathcal{X}/\sqrt{n}$ in the bottleneck distance $d_B$, is a valid confidence set as $\liminf_{n \to \infty} \mathbb{P}\left(\mathcal{P} \in \mathcal{B}_{d_B}(\hat{\mathcal{P}}_h, c_\mathcal{X})\right) \geq 1 - \alpha$. This confidence set has further interpretation that in the persistence diagram, homological features that are above twice the radius $2c_\mathcal{X}$ from the diagonal are simultaneously statistically significant.

## B  Johnson-Lindenstrauss Lemma

Johnson-Lindenstrass Lemma is stated as follows:

**Lemma B.1** (Johnson-Lindenstrauss Lemma). *[34, Lemma 1]*

*Let* $0 < \epsilon < 1$ *and* $\mathcal{X} \subset \mathbb{R}^D$ *be a set of points with size* $n$. *Then for* $d = O\left(\min\left\{n, D, \epsilon^{-2}\log n\right\}\right)$, *there exists a linear map* $f : \mathbb{R}^D \to \mathbb{R}^d$ *such that for all* $x, y \in \mathcal{X}$,

$$(1 - \epsilon)\|x - y\|^2 \le \|f(x) - f(y)\|^2 \le (1 + \epsilon)\|x - y\|^2. \tag{5}$$

*Remark* B.2. This Johnson-Lindenstrass Lemma is optimal for linear maps, and nearly-optimal even if we allow non-linear maps: there exists a set of points $\mathcal{X} \subset \mathbb{R}^D$ with size $n$, such that $d$ should satisfy: (a) $d = \Omega\left(\epsilon^{-2}\log n\right)$ for a linear map $f : \mathbb{R}^D \to \mathbb{R}^d$ satisfying (5) to exist [35, Theorem 3], and (b) $d = \Omega\left(\epsilon^{-2}\log(\epsilon^2 n)\right)$ for a map $f : \mathbb{R}^D \to \mathbb{R}^d$ satisfying (5) to exist [36, Theorem 1].

## C  Denoising topological features from outliers

Using the bootstrap bandwidth $c_{\mathcal{X}}$ as the threshold is the key part of our estimators `TopP&R` for robustly estimating $\mathrm{supp}(P)$. When the level set $\hat{p}_h^{-1}[c_{\mathcal{X}}, \infty)$ is used, the homology of $\hat{p}_h^{-1}[c_{\mathcal{X}}, \infty)$ consists of homological features whose $(\mathrm{birth}) \ge c_{\mathcal{X}}$ and $(\mathrm{death}) \le c_{\mathcal{X}}$, which are the homological features in skyblue area in Figure A1. In this example, we consider three types of homological noise, though there can be many more corresponding to different homological dimensions.

- There can be a $0$-dimensional homological noise of $(\mathrm{birth}) < c_{\mathcal{X}}$ and $(\mathrm{death}) < c_{\mathcal{X}}$, which is the red point in the persistence diagram of Figure A1. This noise corresponds to the orange connected component on the left. As in the figure, this type of homological noise usually corresponds to outliers.

- There can be a $0$-dimensional homological noise of $(\mathrm{birth}) > c_{\mathcal{X}}$ and $(\mathrm{death}) > c_{\mathcal{X}}$, which is the green point in the persistence diagram of Figure A1. This noise corresponds to the connected component surrounded by the green line on the left. As in the figure, this type of homological noise lies within the estimated support, not like the other two.

- There can be a $1$-dimensional homological noise of $(\mathrm{birth}) < c_{\mathcal{X}}$ and $(\mathrm{death}) < c_{\mathcal{X}}$, which is the purple point in the persistence diagram of Figure A1. This noise corresponds to the purple loop on the left.

These homological noises satisfy either their $(\mathrm{birth}) < c_{\mathcal{X}}$ and $(\mathrm{death}) < c_{\mathcal{X}}$ or their $(\mathrm{birth}) > c_{\mathcal{X}}$ and $(\mathrm{death}) > c_{\mathcal{X}}$ simultaneously with high probability, so those homological noises are removed in the estimated support $\hat{p}_h^{-1}[c_{\mathcal{X}}, \infty)$, which is the blue area in the left and the skyblue area in the right in Figure A1.

We would like to further emphasize that homological noises are not restricted to $0$-dimension lying outside the estimated support (red point in the persistence diagram of Figure A1). $0$-dimensional homological noise inside the estimated support (green point in the persistence diagram of Figure A1), $1$-dimensional homological noise can also arise, and the bootstrap bandwidth $c_{\mathcal{X}}$ allows to simultaneously filter them.

## D  Assumptions on distributions and kernels

For distributions, we assume that the order of probability volume decay $P(\mathcal{B}(x, r))$ is at least $r^d$.

**Assumption A1.** *For all* $x \in \mathrm{supp}(P)$ *and* $y \in \mathrm{supp}(Q)$,

$$\liminf_{r \to 0} \frac{P(\mathcal{B}(x, r))}{r^d} > 0, \qquad \liminf_{r \to 0} \frac{Q(\mathcal{B}(y, r))}{r^d} > 0.$$

*Remark* D.1. Assumption A1 is analogous to Assumption 2 of Kim et al. [37], but is weaker since the condition is pointwise on each $x \in \mathbb{R}^d$. And this condition is much weaker than assuming a density on $\mathbb{R}^d$: for example, a distribution supported on a low-dimensional manifold satisfies Assumption A1. This provides a framework suitable for high dimensional data, since many times high dimensional data lies on a low dimensional structure hence its density on $\mathbb{R}^d$ cannot exist. See Kim et al. [37] for a more detailed discussion.

For kernel functions, we assume the following regularity conditions:

**Assumption A2.** *Let* $K : \mathbb{R}^d \to \mathbb{R}$ *be a nonnegative function with* $\|K\|_1 = 1$, $\|K\|_\infty, \|K\|_2 < \infty$, *and satisfy the following:*

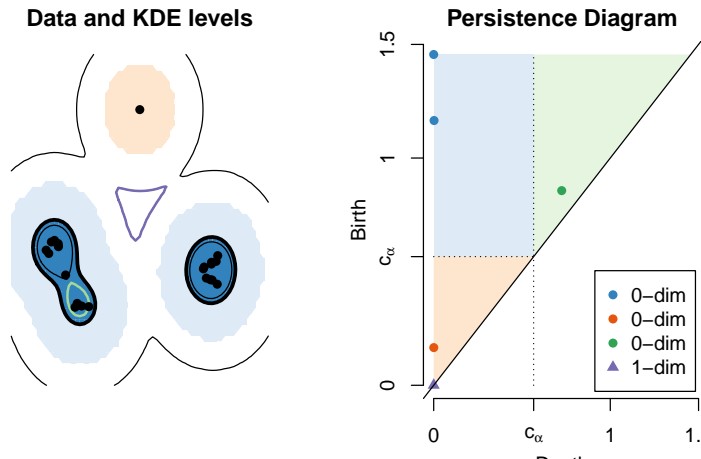

Figure A1: To robustly estimate the support, we use the bootstrap bandwidth $c_\alpha$ to filter out topological noise (orange) and keep topological signal (skyblue). Then `TopP&R` is computed on this support.

*(i)* $K(0) > 0$.

*(ii)* $K$ has a compact support.

*(iii)* $K$ is Lipschitz continuous and of second order.

Assumption A2 allows to build a valid bootstrap confidence band for kernel density estimator (KDE). See Theorem 12 of [26] or Theorem 3.4 of [38]

**Proposition D.2** (Theorem 3.4 of [38])**.** *Let $\mathcal{X} = \{X_1, \ldots, X_n\}$ be IID from a distribution $P$. For $h > 0$, let $\hat{p}_h, \hat{p}_h^*$ be kernel density estimator for $\mathcal{X}$ and its bootstrap $\mathcal{X}^*$, respectively, and for $\alpha \in (0, 1)$, let $c_\mathcal{X}$ be the $\alpha$ bootstrap quantile from $\sqrt{nh^d} \, \|\hat{p}_h - \hat{p}_h^*\|_\infty$. For $h_n \to 0$,*

$$\mathbb{P}\left(\sqrt{nh_n^d} \, \|\hat{p}_{h_n} - p_{h_n}\|_\infty > c_\mathcal{X}\right) = \alpha + \left(\frac{\log n}{nh_n^d}\right)^{\frac{4+d}{4+2d}}.$$

Assumption A1, A2 ensures that, when the bandwidth $h_n \to 0$, average KDEs are bounded away from 0.

**Lemma D.3.** *Let $P$ be a distribution satisfying Assumption A1. Suppose $K$ is a nonnegative function satisfying $K(0) > 0$ and continuous at $0$. Suppose $\{h_n\}_{n \in \mathbb{N}}$ is given with $h_n \geq 0$ and $h_n \to 0$. Then for all $x \in \mathrm{supp}(P)$,*

$$\liminf_{n \to \infty} p_{h_n}(x) > 0.$$

*Proof.* Since $K(0) > 0$ and $K$ is continuous at $0$, there is $r_0 > 0$ such that for all $y \in \mathcal{B}(0, r_0)$, $K(y) \geq \frac{1}{2}K(0) > 0$. And hence

$$p_h(x) = \int \frac{1}{h^d} K\left(\frac{x-y}{h}\right) dP(y) \geq \int \frac{K(0)}{2h^d} 1\left(\frac{x-y}{h} \in \mathcal{B}(0, r_0)\right) dP(y)$$

$$\geq \frac{K(0)}{2h^d} P\left(\mathcal{B}(x, r_0 h)\right).$$

Hence as $h_n \to 0$,

$$\liminf_{n \to \infty} p_{h_n}(x) > 0.$$

$\square$

Before specifying the rate of convergence, we introduce the concept of reach. First introduced by [39], the reach is a quantity expressing the degree of geometric regularity of a set. Given a closed subset $A \subset \mathbb{R}^d$, the medial axis of $A$, denoted by $\mathrm{Med}(A)$, is the subset of $\mathbb{R}^d$ consisting of all the points that have at least two nearest neighbors on $A$.

$$\mathrm{Med}(A) = \left\{ x \in \mathbb{R}^d \setminus A \colon \exists q_1 \neq q_2 \in A, ||q_1 - x|| = ||q_2 - x|| = d(x, A) \right\},$$

where $d(x, A) = \inf_{q \in A} ||q - x||$ denotes the distance from a generic point $x \in \mathbb{R}^d$ to $A$, The reach of $A$ is then defined as the minimal distance from $A$ to $\mathrm{Med}(A)$.

**Definition D.4.** The reach of a closed subset $A \subset \mathbb{R}^d$ is defined as

$$\mathrm{reach}(A) = \inf_{q \in A} d\left(q, \mathrm{Med}(A)\right) = \inf_{q \in A, x \in \mathrm{Med}(A)} ||q - x||.$$

Now, for specifying the rate of convergence, we first assume that distributions have densities away from 0 and $\infty$.

**Assumption A3.** *$P$ and $Q$ have Lebesgue densities $p$ and $q$ that, there exists $0 < p_{\min} \leq p_{\max} < \infty$, $0 < q_{\min} \leq q_{\max} < \infty$ with for all $x \in \mathrm{supp}(P)$ and $y \in \mathrm{supp}(Q)$,*

$$p_{\min} \leq p(x) \leq p_{\max}, \qquad q_{\min} \leq q(x) \leq q_{\max}.$$

We also assume weak geometric assumptions on the support of the distributions $\mathrm{supp}(P)$ and $\mathrm{supp}(Q)$, being bounded and having positive reach.

**Assumption A4.** *We assume $\mathrm{supp}(P)$ and $\mathrm{supp}(Q)$ are bounded. And the support of $P$ and $Q$ have positive reach, i.e. $\mathrm{reach}(\mathrm{supp}(P)) > 0$ and $\mathrm{reach}(\mathrm{supp}(Q)) > 0$.*

Sets with positive reach ensure that the volume of its tubular neighborhood grows in polynomial order, which is Theorem 26 from [40] and originally from [39].

**Proposition D.5.** *Let $A$ be a set with $\mathrm{reach}(A) > 0$. Let $A_r := \{x \in \mathbb{R}^d : d(x, A) \leq r\}$. Then for $r < \mathrm{reach}(A)$, there exists $a_0, \ldots, a_d \in \mathbb{R} \cup \{\infty\}$ that satisfies*

$$\lambda_d(A_r) = \sum_{k=0}^{d-1} a_k r^k,$$

*where $\lambda_d$ is the usual Lebesgue measure of $\mathbb{R}^d$.*

# E    Details and Proofs for Section 4

For a random variable $X$ and $\alpha \in (0, 1)$, let $q_{X,\alpha}$ be upper $\alpha$-quantile of $X$, i.e., $\mathbb{P}(X \geq q_\alpha) = \alpha$, or equivalently, $\mathbb{P}(X < q_\alpha) = 1 - \alpha$. Let $\tilde{p}_h$ be the KDE on $\mathcal{X}^0$. For a finite set $\mathcal{X}$, we use the notation $c_{\mathcal{X},\alpha}$ for $\alpha$-bootstrap quantile satisfying $\mathbb{P}\left( \left\| \hat{p}_{\mathcal{X},h_n} - \hat{p}_{\mathcal{X}^b,h_n} \right\|_\infty < c_{\mathcal{X},\alpha} | \mathcal{X} \right) = 1 - \alpha$, where $\mathcal{X}^b$ is the bootstrap sample from $\mathcal{X}$. For a distribution $P$, we use the notation $c_{P,\alpha}$ for $\alpha$-quantile satisfying $\mathbb{P}\left( \left\| \hat{p}_{h_n} - p_{h_n} \right\|_\infty < c_{P,\alpha} \right) = 1 - \alpha$, where $\hat{p}_{h_n}$ is kernel density estimator of IID samples from $P$. Hence when $\mathcal{X}$ is not IID samples from $P$, the relation of Proposition D.2 may not hold.

*Claim* E.1. Let $Z \sim \mathcal{N}(0, 1)$ be a sample from standard normal distribution, then for $\alpha \in (0, 1)$,

$$q_{Z,\alpha} = \Theta\left( \sqrt{\log(1/\alpha)} \right).$$

And for $0 \leq \delta < \alpha < 1$,

$$q_{Z,\alpha} - q_{Z,\alpha+\delta} = \left( \delta \alpha \sqrt{\log(1/\alpha)} \right)$$

*Proof.* Let $Z \sim \mathcal{N}(0, 1)$, then for all $x > 2$,

$$\frac{1}{2x} \exp\left( -\frac{x^2}{2} \right) \leq \mathbb{P}(Z > x) \leq \frac{1}{x} \exp\left( -\frac{x^2}{2} \right).$$

Then $x = C\sqrt{\log(1/\alpha)}$ gives

$$\frac{\alpha^{C/2}}{2C\sqrt{\log(1/\alpha)}} \leq \mathbb{P}(Z > x) \leq \frac{\alpha^{C/2}}{C\sqrt{\log(1/\alpha)}}.$$

And hence
$$q_{Z,\alpha} = \Theta\left(\sqrt{\log(1/\alpha)}\right),$$
and in particular,
$$\exp\left(-\frac{q_{Z,\alpha}^2}{2}\right) = \Theta\left(\alpha\sqrt{\log(1/\alpha)}\right).$$
Now for $0 \le \delta < \alpha < 1$,
$$(q_{Z,\alpha} - q_{Z,\alpha+\delta})\exp\left(-\frac{q_{Z,\alpha}^2}{2}\right) \le \int_{q_{Z,\alpha+\delta}}^{q_{Z,\alpha}} \exp\left(-\frac{t^2}{2}\right) dt = \delta$$
$$\le (q_{Z,\alpha} - q_{Z,\alpha+\delta})\exp\left(-\frac{q_{Z,\alpha+\delta}^2}{2}\right).$$
And hence
$$q_{Z,\alpha} - q_{Z,\alpha+\delta} \le \delta\exp\left(-\frac{q_{Z,\alpha+\delta}^2}{2}\right) = \Theta\left(\delta(\alpha+\delta)\sqrt{\log(1/\alpha)}\right),$$
and
$$q_{Z,\alpha} - q_{Z,\alpha+\delta} \ge \delta\exp\left(-\frac{q_{Z,\alpha}^2}{2}\right) = \Theta\left(\delta\alpha\sqrt{\log(1/(\alpha+\delta))}\right).$$
Then from $\delta < \alpha$,
$$q_{Z,\alpha} - q_{Z,\alpha+\delta} = \Theta\left(\delta\alpha\sqrt{\log(1/\alpha)}\right).$$

$\square$

*Claim* E.2. Let $X, Y$ be random variables, and $0 \le \delta < \alpha < 1$. Suppose there exists $c > 0$ satisfying
$$\mathbb{P}\left(|X - Y| > c\right) \le \delta. \tag{6}$$
Then,
$$q_{X,\alpha+\delta} - c \le q_{Y,\alpha} \le q_{X,\alpha-\delta} + c.$$

*Proof.* Note that $q_{X,a-\delta}$ satisfies $\mathbb{P}(X > q_{X,\alpha-\delta}) = \alpha - \delta$. Then from (6),
$$\mathbb{P}\left(Y > q_{X,\alpha-\delta} + c\right) \le \mathbb{P}\left(Y > q_{X,\alpha-\delta} + c, |X - Y| \le c\right) + \mathbb{P}\left(|X - Y| > c\right)$$
$$\le \mathbb{P}\left(X > q_{X,\alpha-\delta}\right) + \delta = \alpha.$$
And hence
$$q_{Y,\alpha} \le q_{X,\alpha-\delta} + c.$$
Then changing the role of $X$ and $Y$ gives
$$q_{X,\alpha+\delta} - c \le q_{Y,\alpha}.$$

$\square$

**Lemma E.3.** *(i) Under Assumption 1, 2 and A2,*
$$\|\hat{p}_h - \tilde{p}_h\|_\infty \le \frac{\rho_n \|K\|_\infty}{h^d}.$$

*(ii) Under Assumption 1, 2 and A2, with probability $1 - \delta$,*
$$c_{\mathcal{X}^0,\alpha+\delta} - O\left(\frac{\rho_n}{h^d} + \sqrt{\frac{\rho_n \log(1/\delta)}{nh^{2d}}}\right) \le c_{\mathcal{X},\alpha} \le c_{\mathcal{X}^0,\alpha-\delta} + O\left(\frac{\rho_n}{h^d} + \sqrt{\frac{\rho_n \log(1/\delta)}{nh^{2d}}}\right).$$

*(iii) Suppose Assumption 1, 2, A2 hold, and let $\alpha, \delta_n \in (0, 1)$. Suppose $nh_n^d \to \infty$, $\delta_n^{-1} = O\left(\left(\frac{\log n}{nh_n^d}\right)^{\frac{4+d}{4+2d}}\right)$ and $nh_n^{-d}\rho_n^2\delta_n^{-2} \to 0$. Then with probability $1 - \alpha - 4\delta_n$,*
$$\|\hat{p}_h - p_h\|_\infty < c_{\mathcal{X},\alpha} \le c_{P,\alpha-3\delta_n}.$$

*Proof.* (i)

First, note that

$$\hat{p}_h(x) - \tilde{p}_h(x) = \frac{1}{nh^d} \sum_{i=1}^{n} \left( K\left(\frac{x - X_i}{h}\right) - K\left(\frac{x - X_i^0}{h}\right) \right).$$

Then under Assumption A2,

$$\|\hat{p}_h - \tilde{p}_h\|_\infty \leq \frac{1}{nh^d} \sum_{i=1}^{n} \left\| K\left(\frac{\cdot - X_i}{h}\right) - K\left(\frac{\cdot - X_i^0}{h}\right) \right\|_\infty$$

$$\leq \frac{1}{nh^d} \sum_{i=1}^{n} \|K\|_\infty \, I\left(X_i \neq X_i^0\right).$$

Then from Assumption 2, $\sum_{i=1}^{n} I\left(X_i \neq X_i^0\right) \leq n\rho_n$, and hence

$$\|\hat{p}_h - \tilde{p}_h\|_\infty \leq \frac{\|K\|_\infty \, \rho_n}{h^d}.$$

(ii)

Let $\mathcal{X}_b$, $\mathcal{X}_b^0$ be bootstrapped samples of $\mathcal{X}$, $\mathcal{X}^0$ with the same sampling with replacement process. Let $\hat{p}_h^b, \tilde{p}_h^b$ be KDE of $\mathcal{X}_b$ and $\mathcal{X}_b^0$, respectively. And, note that

$$\left| \left\| \hat{p}_h - \hat{p}_h^b \right\|_\infty - \left\| \tilde{p}_h - \tilde{p}_h^b \right\|_\infty \right| \leq \|\hat{p}_h - \tilde{p}_h\|_\infty + \left\| \hat{p}_h^b - \tilde{p}_h^b \right\|_\infty.$$

Let $L_b$ be the number of elements where $\mathcal{X}_b$ and $\mathcal{X}_b^0$ differ, i.e., $L_b = \left| \mathcal{X}_b \backslash \mathcal{X}_b^0 \right| = \left| \mathcal{X}_b^0 \backslash \mathcal{X}_b \right|$, then $L_b \sim \text{Binomial}(n, \rho_n)$, and

$$\left\| \hat{p}_h^b - \tilde{p}_h^b \right\|_\infty \leq \frac{\|K\|_\infty \, L_b}{nh^d}.$$

And hence,

$$\left| \left\| \hat{p}_h - \hat{p}_h^b \right\|_\infty - \left\| \tilde{p}_h - \tilde{p}_h^b \right\|_\infty \right| \leq \frac{\|K\|_\infty \, (n\rho_n + L_b)}{nh^d}. \tag{7}$$

Then by Hoeffding's inequality, with probability $1 - \delta$,

$$L_b \leq n\rho_n + \sqrt{\frac{n \log(1/\delta)}{2}}.$$

by applying this to (7), with probability $1 - \delta$,

$$\left| \left\| \hat{p}_h - \hat{p}_h^b \right\|_\infty - \left\| \tilde{p}_h - \tilde{p}_h^b \right\|_\infty \right| \leq O\left( \frac{\rho_n}{h^d} + \sqrt{\frac{\rho_n \log(1/\delta)}{nh^{2d}}} \right).$$

Hence applying Claim E.2 'implies that, with probability $1 - \delta$,

$$c_{\mathcal{X}^0, \alpha + \delta} - O\left( \frac{\rho_n}{h^d} + \sqrt{\frac{\rho_n \log(1/\delta)}{nh^{2d}}} \right) \leq c_{\mathcal{X}, \alpha} \leq c_{\mathcal{X}^0, \alpha - \delta} + O\left( \frac{\rho_n}{h^d} + \sqrt{\frac{\rho_n \log(1/\delta)}{nh^{2d}}} \right).$$

(iii)

Let $\tilde{\delta}_n := O\left( \left( \frac{\log n}{nh_n^d} \right)^{\frac{4+d}{4+2d}} \right)$ be from RHS of Proposition D.2, then for large enough $n$, $\delta_n \geq \tilde{\delta}_n$. Note that Proposition D.2 implies that for all $\alpha \in (0, 1)$,

$$c_{P, \alpha + \delta_n} \leq c_{\mathcal{X}_0, \alpha} \leq c_{P, \alpha - \delta_n}. \tag{8}$$

Now, $nh_n^d \to \infty$ implies that $\sqrt{nh_n^d}(\tilde{p}_{h_n} - p_{h_n})$ converges to a Gaussian process, and then Claim E.2 implies that $c_{P, \alpha} = \Theta\left( \sqrt{\frac{\log(1/\alpha)}{nh_n^d}} \right)$ and

$$c_{P,\alpha} - c_{P,\alpha+\delta_n} = \Theta\left(\delta_n\alpha\sqrt{\frac{\log(1/\alpha)}{nh_n^d}}\right). \tag{9}$$

Then under Assumption 2, since $nh_n^{-d}\rho_n^2\delta_n^{-2} = o(1)$ and $n\rho_n \geq 1$ implies $h_n^{-d}\rho_n\delta_n^{-2} = o(1)$, and then

$$\frac{\rho_n}{h_n^d} + \sqrt{\frac{\rho_n\log(1/\delta)}{nh_n^{2d}}} = O\left(\delta_n\alpha\sqrt{\frac{\log(1/\alpha)}{nh_n^d}}\right). \tag{10}$$

Then (8), (9), (10) implies that

$$c_{P,\alpha+3\delta_n} \leq c_{P,\alpha+2\delta_n} - O\left(\delta_n\alpha\sqrt{\frac{\log(1/\alpha)}{nh_n^d}}\right) \leq c_{\mathcal{X}^0,\alpha+\delta_n} - O\left(\frac{\rho_n}{h_n^d} + \sqrt{\frac{\rho_n\log(1/\delta)}{nh_n^{2d}}}\right), \tag{11}$$

and

$$c_{\mathcal{X}^0,\alpha-\delta_n} + O\left(\frac{\rho_n}{h_n^d} + \sqrt{\frac{\rho_n\log(1/\delta)}{nh_n^{2d}}}\right) \leq c_{P,\alpha-2\delta_n} + O\left(\delta_n\alpha\sqrt{\frac{\log(1/\alpha)}{nh_n^d}}\right) \leq c_{P,\alpha-3\delta_n}. \tag{12}$$

Now, from the definition of $c_{P,\alpha+3\delta_n}$, with probability $1 - \alpha - 3\delta_n$,

$$\|\hat{p}_h - p_h\|_\infty \leq c_{P,\alpha+3\delta_n}. \tag{13}$$

And (ii) implies that, with probability $1 - \delta_n$,

$$c_{\mathcal{X}^0,\alpha+\delta_n} - O\left(\frac{\rho_n}{h_n^d} + \sqrt{\frac{\rho_n\log(1/\delta)}{nh_n^{2d}}}\right) \leq c_{\mathcal{X},\alpha} \leq c_{\mathcal{X}^0,\alpha-\delta_n} + O\left(\frac{\rho_n}{h_n^d} + \sqrt{\frac{\rho_n\log(1/\delta)}{nh_n^{2d}}}\right). \tag{14}$$

Hence by combining (11), (12), (13), (14), with probability $1 - \alpha - 4\delta_n$,

$$\|\hat{p}_h - p_h\|_\infty \leq c_{P,\alpha+3\delta_n}$$

$$\leq c_{\mathcal{X}^0,\alpha+\delta_n} - O\left(\frac{\rho_n}{h_n^d} + \sqrt{\frac{\rho_n\log(1/\delta)}{nh_n^{2d}}}\right)$$

$$\leq c_{\mathcal{X},\alpha}$$

$$\leq c_{\mathcal{X}^0,\alpha-\delta_n} + O\left(\frac{\rho_n}{h_n^d} + \sqrt{\frac{\rho_n\log(1/\delta)}{nh_n^{2d}}}\right)$$

$$\leq c_{P,\alpha-3\delta_n}.$$

$\square$

**Corollary E.4.** *Suppose Assumption 1, 2, A2 hold, and let $\alpha \in (0,1)$. Suppose $\delta_n^{-1} = O\left(\left(\frac{\log n}{nh_n^d}\right)^{\frac{4+d}{4+2d}}\right)$ and $nh_n^{-d}\rho_n^2\delta_n^{-2} \to 0$. Then with probability $1 - \alpha - 4\delta_n$,*

*(i) Let $\delta_n \in (0,1)$ and suppose $nh_n^d \to \infty$, $\delta_n^{-1} = O\left(\left(\frac{\log n}{nh_n^d}\right)^{\frac{4+d}{4+2d}}\right)$ and $nh_n^{-d}\rho_n^2\delta_n^{-2} \to 0$. Then with probability $1 - \alpha - 4\delta_n$,*

$$p_{h_n}^{-1}[2c_{P,\alpha-3\delta_n}, \infty) \subset \hat{p}_{h_n}^{-1}[c_{\mathcal{X},\alpha}, \infty) \subset \text{supp}(P_{h_n}).$$

*(ii) Let $\delta_m \in (0,1)$ and suppose $mh_m^d \to \infty$, $\delta_m^{-1} = O\left(\left(\frac{\log m}{mh_m^d}\right)^{\frac{4+d}{4+2d}}\right)$ and $mh_m^{-d}\rho_m^2\delta_m^{-2} \to 0$. Then with probability $1 - \alpha - 4\delta_m$,*

$$q_{h_m}^{-1}[2c_{Q,\alpha-3\delta_m}, \infty) \subset \hat{q}_{h_m}^{-1}[c_{\mathcal{Y},\alpha}, \infty) \subset \text{supp}(Q_{h_m}).$$

*Proof.* (i)

Lemma E.3 (iii) implies that with probability $1 - \alpha - 4\delta_n$, $\|\hat{p}_h - p_h\| < c_{\mathcal{X},\alpha} \le c_{P,\alpha-3\delta_n}$. This implies

$$p_{h_n}^{-1}[2c_{P,\alpha-3\delta_n}, \infty) \subset \hat{p}_{h_n}^{-1}[c_{\mathcal{X},\alpha}, \infty) \subset \text{supp}(P_{h_n}).$$

(ii)

This can be proven similarly to (i).

$\square$

*Claim* E.5. For a nonnegative measure $\mu$ and sets $A, B, C, D$,
$$\mu(A \cap B) - \mu(C \cap D) \le \mu(A\backslash C) + \mu(B\backslash D).$$

*Proof.*

$$\begin{aligned}
\mu(A \cap B) - \mu(C \cap D) &\le \mu((A \cap B)\backslash(C \cap D)) = \mu((A \cap B) \cap (C^{\complement} \cup D^{\complement})) \\
&= \mu((A \cap B) \cap C^{\complement}) \cup (A \cap B) \cap D^{\complement})) \\
&\le \mu((A \cap B)\backslash C) + \mu(A \cap B)\backslash D) \\
&\le \mu(A\backslash C) + \mu(B\backslash D).
\end{aligned}$$

$\square$

From here, let $P_n$ and $Q_m$ be the empirical measures on $\mathcal{X}$ and $\mathcal{Y}$, respectively, i.e., $P_n = \frac{1}{n}\sum_{i=1}^n \delta_{X_i}$ and $Q_m = \frac{1}{m}\sum_{j=1}^m \delta_{Y_j}$.

**Lemma E.6.** *Suppose Assumption 1, 2 hold. Let $A \subset \mathbb{R}^d$. Then with probability $1 - \delta$,*

$$|P_n - P|(A) \le \rho_n + \sqrt{\frac{\log(2/\delta)}{2n}},$$

*and in particular,*

$$P_n(A) \le P(A) + \rho_n + \sqrt{\frac{\log(2/\delta)}{n}}.$$

*Proof.* Let $P_n^0$ be the empirical measure on $\mathcal{X}^0$, i.e., $P_n^0 = \frac{1}{n}\sum_{i=1}^n \delta_{X_i^0}$. By using Hoeffding's inequality,

$$\mathbb{P}\left(\left|(P_n^0 - P)(A)\right| \ge t\right) \le 2\exp\left(-2nt^2\right),$$

and hence with probability $1 - \delta$,

$$\left|P_n^0 - P\right|(A) \le \sqrt{\frac{\log(2/\delta)}{2n}}.$$

And $\left|P_n - P_n^0\right|(A)$ is expanded as

$$\left|P_n - P_n^0\right|(A) = \frac{1}{n}\sum_{i=1}^n \left|I(X_i \in A) - I(X_i^0 \in A)\right|.$$

Under Assumption 2, $\sum_{i=1}^n I\left(X_i \ne X_i^0\right) \le n\rho_n$, and hence

$$\begin{aligned}
\left|P_n - P_n^0\right|(A) &= \frac{1}{n}\sum_{i=1}^n \left|I(X_i \in A) - I(X_i^0 \in A)\right| \\
&\le \frac{1}{n}\sum_{i=1}^n I\left(X_i \ne X_i^0\right) = \rho_n.
\end{aligned}$$

Therefore, with probability $1 - \delta$,

$$|P_n - P|(A) \le \rho_n + \sqrt{\frac{\log(2/\delta)}{2n}}.$$

$\square$

*Claim* E.7. Suppose Assumption 1, 2, A1, A2 hold, and let $\delta_n, \delta_m \in (0,1)$. Suppose $nh_n^d \to \infty$, $mh_m^d \to \infty$, $\delta_n^{-1} = O\left(\left(\frac{\log n}{nh_n^d}\right)^{\frac{4+d}{4+2d}}\right)$, $\delta_m^{-1} = O\left(\left(\frac{\log m}{mh_m^d}\right)^{\frac{4+d}{4+2d}}\right)$, $nh_n^{-d}\rho_n^2\delta_n^{-2} \to 0$, and $nh_m^{-d}\rho_m^2\delta_m^{-2} \to 0$. Let

$$B_{n,m} := \left(\operatorname{supp}(P)\backslash p_{h_n}^{-1}[2c_P, \infty)\right) \cup q_{h_m}^{-1}(0, 2c_Q) \cup \operatorname{supp}(P_{h_n})\backslash\operatorname{supp}(P). \tag{15}$$

(i) With probability $1 - 2\alpha - 4\delta_n - 8\delta_m$,

$$\left|Q_m\left(\hat{p}_{h_n}^{-1}[c_{\mathcal{X}}, \infty) \cap \hat{q}_{h_m}^{-1}[c_{\mathcal{Y}}, \infty)\right) - Q_m\left(\operatorname{supp}(P) \cap \operatorname{supp}(Q)\right)\right|$$
$$\leq C\left(Q(B_{n,m}) + \rho_m + \sqrt{\frac{\log(1/\delta)}{m}}\right).$$

(ii) With probability $1 - 2\alpha - 8\delta_m$,

$$\left|Q_m\left(\hat{q}_{h_m}^{-1}[c_{\mathcal{Y}}, \infty)\right) - Q_m\left(\operatorname{supp}(Q)\right)\right| \leq C\left(Q\left(q_{h_m}^{-1}(0, 2c_Q)\right) + \rho_m + \sqrt{\frac{\log(1/\delta)}{m}}\right).$$

(iii) With probability $1 - 2\alpha - 4\delta_n - 9\delta_m$,

$$\left|Q_m\left(\hat{p}_{h_n}^{-1}[c_{\mathcal{X}}, \infty) \cap \hat{q}_{h_m}^{-1}[c_{\mathcal{Y}}, \infty)\right) - Q\left(\operatorname{supp}(P)\right)\right|$$
$$\leq C\left(Q(B_{n,m}) + \rho_m + \sqrt{\frac{\log(1/\delta)}{m}}\right).$$

(iv) With probability $1 - 2\alpha - 9\delta_m$,

$$\left|Q_m\left(\hat{q}_{h_m}^{-1}[c_{\mathcal{Y}}, \infty)\right) - 1\right| \leq C\left(Q\left(q_{h_m}^{-1}(0, 2c_Q)\right) + \rho_m + \sqrt{\frac{\log(1/\delta)}{m}}\right).$$

(v) As $n, m \to \infty$, $B_{n,m} \to \emptyset$. And in particular,

$$Q(B_{n,m}) \to 0.$$

*Proof.* (i)

From Corollary E.4 (i) and (ii), with probability $1 - 2\alpha - 4(\delta_n + \delta_m)$,

$$Q_m\left(p_{h_n}^{-1}[2c_P, \infty) \cap q_{h_m}^{-1}[2c_Q, \infty)\right) \leq Q_m\left(\hat{p}_{h_n}^{-1}[c_{\mathcal{X}}, \infty) \cap \hat{q}_{h_m}^{-1}[c_{\mathcal{Y}}, \infty)\right)$$
$$\leq Q_m\left(\operatorname{supp}(P_{h_n}) \cap \operatorname{supp}(Q_{h_m})\right). \tag{16}$$

Then from the first inequality of (16), combining with Claim E.5 gives

$$Q_m\left(\hat{p}_{h_n}^{-1}[c_{\mathcal{X}}, \infty) \cap \hat{q}_{h_m}^{-1}[c_{\mathcal{Y}}, \infty)\right) - Q_m\left(\operatorname{supp}(P) \cap \operatorname{supp}(Q)\right)$$
$$\geq Q_m\left(p_{h_n}^{-1}[2c_P, \infty) \cap q_{h_m}^{-1}[2c_Q, \infty)\right) - Q_m\left(\operatorname{supp}(P) \cap \operatorname{supp}(Q)\right)$$
$$\geq -\left(Q_m\left(\operatorname{supp}(P)\backslash p_{h_n}^{-1}[2c_P, \infty)\right) + Q_m\left(\operatorname{supp}(Q)\backslash q_{h_m}^{-1}[2c_Q, \infty)\right)\right). \tag{17}$$

And from the second inequality of (16), combining with Claim E.5 gives

$$Q_m\left(\hat{p}_{h_n}^{-1}[c_{\mathcal{X}}, \infty) \cap \hat{q}_{h_m}^{-1}[c_{\mathcal{Y}}, \infty)\right) - Q_m\left(\operatorname{supp}(P) \cap \operatorname{supp}(Q)\right)$$
$$\leq Q_m\left(\operatorname{supp}(P_{h_n}) \cap \operatorname{supp}(Q_{h_m})\right) - Q_m\left(\operatorname{supp}(P) \cap \operatorname{supp}(Q)\right)$$
$$\leq Q_m\left(\operatorname{supp}(P_{h_n})\backslash\operatorname{supp}(P)\right) + Q_m\left(\operatorname{supp}(Q_{h_m})\backslash\operatorname{supp}(Q)\right). \tag{18}$$

And hence combining (17) and (18) gives that, with probability $1 - 2\alpha - 4(\delta_n + \delta_m)$,

$$\left|Q_m\left(\hat{p}_{h_n}^{-1}[c_{\mathcal{X}}, \infty) \cap \hat{q}_{h_m}^{-1}[c_{\mathcal{Y}}, \infty)\right) - Q_m\left(\operatorname{supp}(P) \cap \operatorname{supp}(Q)\right)\right|$$
$$\leq \max\left\{Q_m\left(\operatorname{supp}(P)\backslash p_{h_n}^{-1}[2c_P, \infty)\right) + Q_m\left(\operatorname{supp}(Q)\backslash q_{h_m}^{-1}[2c_Q, \infty)\right)\right.$$
$$\left., Q_m\left(\operatorname{supp}(P_{h_n})\backslash\operatorname{supp}(P)\right) + Q_m\left(\operatorname{supp}(Q_{h_m})\backslash\operatorname{supp}(Q)\right)\right\}. \tag{19}$$

Now we further bound the upper bound of (19). From Lemma E.6, with probability $1 - \delta_m$,

$$Q_m \left( \text{supp}(P) \backslash p_{h_n}^{-1}[2c_P, \infty) \right) \leq Q \left( \text{supp}(P) \backslash p_{h_n}^{-1}[2c_P, \infty) \right) + \rho_m + \sqrt{\frac{\log(2/\delta)}{2m}}. \quad (20)$$

And similarly, with probability $1 - \delta_m$,

$$Q_m \left( \text{supp}(Q) \backslash q_{h_m}^{-1}[2c_Q, \infty) \right) \leq Q \left( \text{supp}(Q) \backslash q_{h_m}^{-1}[2c_Q, \infty) \right) + \rho_m + \sqrt{\frac{\log(2/\delta)}{2m}}$$

$$= Q \left( q_{h_m}^{-1}(0, \infty) \right) + \rho_m + \sqrt{\frac{\log(2/\delta)}{2m}}. \quad (21)$$

And similarly, with probability $1 - \delta_m$,

$$Q_m \left( \text{supp}(P_{h_n}) \backslash \text{supp}(P) \right) \leq Q \left( \text{supp}(P_{h_n}) \backslash \text{supp}(P) \right) + \rho_m + \sqrt{\frac{\log(2/\delta)}{2m}}. \quad (22)$$

And similarly, with probability $1 - \delta_m$,

$$Q_m \left( \text{supp}(Q_{h_m}) \backslash \text{supp}(Q) \right) \leq Q \left( \text{supp}(Q_{h_m}) \backslash \text{supp}(Q) \right) + \rho_m + \sqrt{\frac{\log(2/\delta)}{2m}}$$

$$= \rho_m + \sqrt{\frac{\log(2/\delta)}{2m}}. \quad (23)$$

Hence by applying (20), (21), (22), (23) to (19), with probability $1 - 2\alpha - 4\delta_n - 8\delta_m$,

$$\left| Q_m \left( \hat{p}_{h_n}^{-1}[c_{\mathcal{X}}, \infty) \cap \hat{q}_{h_m}^{-1}[c_{\mathcal{Y}}, \infty) \right) - Q_m \left( \text{supp}(P) \cap \text{supp}(Q) \right) \right|$$

$$\leq C \left( Q \left( \left( \text{supp}(P) \backslash p_{h_n}^{-1}[2c_P, \infty) \right) \cup q_{h_m}^{-1}(0, 2c_Q) \cup \text{supp}(P_{h_m}) \backslash \text{supp}(P) \right) \right.$$

$$\left. + \rho_m + \sqrt{\frac{\log(1/\delta)}{m}} \right)$$

$$= C \left( Q(B_{n,m}) + \rho_m + \sqrt{\frac{\log(1/\delta)}{m}} \right),$$

where $B_{n,m}$ is from (15).

(ii)

This can be done similarly to (i).

(iii)

Lemma E.6 gives that with probability $1 - \delta_m$,

$$\left| Q_m \left( \text{supp}(P) \cap \text{supp}(Q) \right) - Q \left( \text{supp}(P) \right) \right| \leq \rho_m + \sqrt{\frac{\log(2/\delta)}{2m}}.$$

Hence combining this with (i) gives the desired result.

(iv)

Lemma E.6 gives that with probability $1 - \delta_m$,

$$\left| Q_m \left( \text{supp}(Q) \right) - 1 \right| \leq \rho_m + \sqrt{\frac{\log(2/\delta)}{2m}}.$$

Hence combining this with (ii) gives the desired result.

(v)

Note that Lemma D.3 implies that for all $x \in \text{supp}(P)$, $\liminf_{n \to \infty} p_{h_n}(x) > 0$, so $p_{h_n}(x) > 2c_P$ for large enough $n$. And hence as $n \to \infty$,

$$\text{supp}(P) \backslash p_{h_n}^{-1}[2c_P, \infty) \to \emptyset. \quad (24)$$

And similar argument holds for $\mathrm{supp}(Q)\backslash q_{h_m}^{-1}[2c_Q, \infty)$, so as $m \to \infty$,

$$\mathrm{supp}(Q)\backslash q_{h_m}^{-1}[2c_Q, \infty) \to \emptyset. \tag{25}$$

Also, since $K$ has compact support, for any $x \notin \mathrm{supp}(P)$, $x \notin \mathrm{supp}(P_{h_n})$ once $h_n < d(x, \mathrm{supp}(P))$. Hence as $n \to \infty$,

$$\mathrm{supp}(P_{h_n})\backslash\mathrm{supp}(P) \to \emptyset. \tag{26}$$

And similarly, as $m \to \infty$,

$$\mathrm{supp}(Q_{h_m})\backslash\mathrm{supp}(Q) \to \emptyset. \tag{27}$$

Hence by applying (24), (25), (26), (27) to the definition of $B_{n,m}$ in (15) gives that as $n, m \to \infty$,

$$B_{n,m} \to \emptyset.$$

And in particular,

$$Q(B_{n,m}) \to 0.$$

$\square$

Below we state a more formal version of Proposition 4.1.

**Proposition E.8.** *Suppose Assumption 1,2,A1,A2 hold. Suppose $\alpha \to 0$, $\delta_n \to 0$, $\delta_n^{-1} = O\left(\left(\frac{\log n}{nh_n^d}\right)^{\frac{4+d}{4+2d}}\right)$, $h_n \to 0$, $nh_n^d \to \infty$, and $nh_n^{-d}\rho_n^2\delta_n^{-2} \to 0$, and similar relations hold for $h_m$, $\rho_m$. Then there exists some constant $C > 0$ not depending on anything else such that*

$$\left|\mathrm{TopP}_{\mathcal{X}}(\mathcal{Y}) - \mathrm{precision}_P(\mathcal{Y})\right| \le C\left(Q(B_{n,m}) + \rho_m + \sqrt{\frac{\log(1/\delta)}{m}}\right),$$

$$\left|\mathrm{TopR}_{\mathcal{Y}}(\mathcal{X}) - \mathrm{recall}_Q(\mathcal{X})\right| \le C\left(P(A_{n,m}) + \rho_n + \sqrt{\frac{\log(1/\delta)}{n}}\right),$$

*where*

$$A_{n,m} := \left(\mathrm{supp}(Q)\backslash q_{h_m}^{-1}[2c_Q, \infty)\right) \cup p_{h_n}^{-1}(0, 2c_P) \cup \mathrm{supp}(Q_{h_m})\backslash\mathrm{supp}(Q),$$
$$B_{n,m} := \left(\mathrm{supp}(P)\backslash p_{h_n}^{-1}[2c_P, \infty)\right) \cup q_{h_m}^{-1}(0, 2c_Q) \cup \mathrm{supp}(P_{h_n})\backslash\mathrm{supp}(P).$$

*And as $n, m \to \infty$, $P(A_{n,m}) \to 0$ and $Q(B_{n,m}) \to 0$ hold.*

*Proof of Proposition E.8.* Now this is an application of Claim E.7 (i) (ii) (v) to the definitions of $\mathrm{precision}_P(\mathcal{Y})$ and $\mathrm{recall}_Q(\mathcal{X})$ in (1) and (2) and the definitions of $\mathrm{TopP}_{\mathcal{X}}(\mathcal{Y})$ and $\mathrm{TopR}_{\mathcal{Y}}(\mathcal{X})$ in (3) and (4).

$\square$

Similarly, we state a more formal version of Theorem 4.2.

**Theorem E.9.** *Suppose Assumption 1,2,A1,A2 hold. Suppose $\alpha \to 0$, $\delta_n \to 0$, $\delta_n^{-1} = O\left(\left(\frac{\log n}{nh_n^d}\right)^{\frac{4+d}{4+2d}}\right)$, $h_n \to 0$, $nh_n^d \to \infty$, and $nh_n^{-d}\rho_n^2\delta_n^{-2} \to 0$, and similar relations hold for $h_m$, $\rho_m$. Then there exists some constant $C > 0$ not depending on anything else such that*

$$\left|\mathrm{TopP}_{\mathcal{X}}(\mathcal{Y}) - \mathrm{precision}_P(Q)\right| \le C\left(Q(B_{n,m}) + \rho_m + \sqrt{\frac{\log(1/\delta)}{m}}\right),$$

$$\left|\mathrm{TopR}_{\mathcal{Y}}(\mathcal{X}) - \mathrm{recall}_Q(P)\right| \le C\left(P(A_{n,m}) + \rho_n + \sqrt{\frac{\log(1/\delta)}{n}}\right),$$

*where $A_{n,m}$ and $B_{n,m}$ are the same as in Proposition E.8. Again as $n, m \to \infty$, $P(A_{n,m}) \to 0$ and $Q(B_{n,m}) \to 0$ hold.*

*Proof of Theorem E.9.* Now this is an application of Claim E.7 (iii) (iv) (v) to the definitions of $\text{precision}_P(Q)$ and $\text{recall}_Q(P)$ and the definitions of $\text{TopP}_{\mathcal{X}}(\mathcal{Y})$ and $\text{TopR}_{\mathcal{Y}}(\mathcal{X})$ in (3) and (4).

$\square$

*Proof of Lemma 4.3.* Recall that $B_{n,m}$ is defined in (15) as

$$B_{n,m} = \left(\text{supp}(P)\backslash p_{h_n}^{-1}[2c_P, \infty)\right) \cup q_{h_m}^{-1}(0, 2c_Q) \cup \text{supp}(P_{h_n})\backslash\text{supp}(P),$$

and hence $Q(B_{n,m})$ can be upper bounded as

$$Q(B_{n,m}) \leq Q\left(\text{supp}(P)\backslash p_{h_n}^{-1}[2c_P, \infty)\right) + Q\left(q_{h_m}^{-1}(0, 2c_Q)\right) + Q\left(\text{supp}(P_{h_n})\backslash\text{supp}(P)\right). \quad (28)$$

For the first term of RHS of (28), suppose $\text{supp}(K) \subset \mathcal{B}_{\mathbb{R}^d}(0,1)$ for convenience, and let $A_{-h_n} := \left\{x \in \text{supp}(P) : d(x, \mathbb{R}^d\backslash\text{supp}(P)) \geq h_n\right\}$. Then from Assumption A3, for all $x \in A_{-h_n}$, $p_{h_n}(x) \geq p_{\min}$ holds. Hence for large enough $N$ such that for $n \geq N$, $c_P < p_{\min}$, then $p_{h_n}(x) \geq p_{\min}$ for all $x \in A_{-h_n}$, and hence

$$Q\left(\text{supp}(P)\backslash p_{h_n}^{-1}[2c_P, \infty)\right) \leq Q\left(\text{supp}(P)\backslash A_{-h_n}\right). \quad (29)$$

Also, $\text{supp}(P)$ being bounded implies that all the coefficients $a_0, \ldots, a_{d-1}$ in Proposition D.5 for $\text{supp}(P)$ are in fact finite. Then $q \leq q_{\max}$ from Assumption A3 and Proposition D.5 implies

$$Q\left(\text{supp}(P)\backslash A_{-h_n}\right) = O(h_n). \quad (30)$$

And hence combining (29) and (30) gives that

$$Q\left(\text{supp}(P)\backslash p_{h_n}^{-1}[2c_P, \infty)\right) = O(h_n). \quad (31)$$

For the second term of RHS of (28), with a similar argument,

$$Q\left(q_{h_m}^{-1}(0, 2c_Q)\right) = O(h_m). \quad (32)$$

Finally, for the third term of RHS of (28), Proposition D.5 implies

$$Q\left(\text{supp}(P_{h_n})\backslash\text{supp}(P)\right) = O(h_n). \quad (33)$$

Hence applying (31), (32), (33) to (28) gives that

$$Q(B_{n,m}) = O(h_n + h_m).$$

$\square$

# F   Related Work

**Improved Precision & Recall** (P&R). Existing metrics such as IS and FID assess the performance of generative models with a single score while showing usefulness in determining performance rankings between models and are still widely used. These evaluation metrics have problems that they cannot provide detailed interpretations of the evaluation in terms of fidelity and diversity. Sajjadi et al. [12] tried to solve this problem by introducing the original Precision and Recall, which however has a limitation as it is inaccurate due to the simultaneous approximation of real support and fake support through "$k$-means clustering". For example, if we evaluate the fake images having high fidelity and large value of $k$, many fake features can be classified in a small cluster with no real features, resulting in a low fidelity score ($precision_P := Q(supp(P))$). Kynkäänniemi et al. [13] focuses on the limitation from the support estimation and presents an P&R that allows us to more accurately assess fidelity and diversity by approximating real support and fake support separately:

$$precision(\mathcal{X}, \mathcal{Y}) := \frac{1}{M}\sum_{i}^{M} f(Y_i, \mathcal{X}), \; recall(\mathcal{X}, \mathcal{Y}) := \frac{1}{N}\sum_{j}^{N} f(X_j, \mathcal{Y}),$$

$$\text{where } f(Y_i, \mathcal{X}) = \begin{cases} 1, & \text{if } Y_i \in supp(P), \\ 0, & \text{otherwise.} \end{cases}$$

Here, $supp(P)$ is defined as the union of spheres centered on each real feature $X_i$ and whose radius is the distance between $k$th nearest real features of $X_i$

**Density & Coverage** (D&C). Naeem et al. [14] has reported that P&R cannot stably provide accurate fidelity and diversity when real or fake features are present through experiment. This limitation of P&R is that it has a problem with approximating overestimated supports by outliers due to the use of "$k$-nearest neighborhood" algorithm. For a metric that is robust to outlying features, Naeem et al. [14] proposes a new evaluation metric that only relies on the real support, based on the fact that a generative model often generates artifacts which possibly results in outlying features in the embedding space:

$$density(\mathcal{X}, \mathcal{Y}) := \frac{1}{M} \sum_{j}^{M} \sum_{i}^{N} 1_{Y_j \in f(X_i)}, \; coverage(\mathcal{X}, \mathcal{Y}) := \frac{1}{N} \sum_{i}^{N} 1_{\exists j \; s.t. \; Y_j \in f(X_i)},$$

$$\text{where } f(X_i) = \mathcal{B}(X_i, NND_k(X_i))$$

In the equation, $NND_k(X_i)$ means the $k$th-nearest neighborhood distance of $X_i$ and $\mathcal{B}(X_i, NND_k(X_i))$ denotes the hyper-sphere centered at $X_i$ with radius $NND_k(X_i)$. However, D&C is only a partial solution because it still gives an inaccurate evaluation when a real outlying feature exists. Unlike coverage, density is not upper-bounded which makes it unclear exactly which ideal score a generative model should achieve.

**Geometric Evaluation of Data Representations** (GCA). Poklukar et al. [27] proposes a metric called GCM that assesses the fidelity and diversity of fake images. GCM uses the geometric and topological properties of connected components formed by vertex $\mathcal{V}$ (feature) and edge $\mathcal{E}$ (connection). Briefly, when the pairwise distance between vertices is less than a certain threshold $\epsilon$, a connected component is formed by connecting two vertices with an edge. This set of connected components can be thought of as a graph $\mathcal{G} = (\mathcal{V}, \mathcal{E})$, and each connected component separated from each other in $\mathcal{G}$ is defined as a subgraph $\mathcal{G}_i$ (i.e., $\mathcal{G} = \cup_i \mathcal{G}_i$). GCA uses the consistency $c(\mathcal{G}_i)$ and the quality $q(\mathcal{G}_i)$ to select the important subgraph $\mathcal{G}_i$ that is formed on both real vertices and fake vertices ($\mathcal{V} = \mathcal{X} \cup \mathcal{Y}$). $c(\mathcal{G}_i)$ evaluates the ratio of the number of real vertices and fake vertices and $q(\mathcal{G}_i)$ computes the ratio of the number of edges connecting real vertices and fake vertices to the total number of edges in $\mathcal{G}_i$. Given the consistency threshold $\eta_c$ and quality threshold $\eta_q$, the set of important subgraphs is defined as $\mathcal{S}(\eta_c, \eta_q) = \cup_{q(\mathcal{G}_i) > \eta_q, \; c(\mathcal{G}_i) > \eta_c} \mathcal{G}_i$. Based on this, GCA precision and recall are defined as follows:

$$\text{GCA precision} = \frac{|\mathcal{S}^{\mathcal{Y}}|_{\mathcal{V}}}{|\mathcal{G}^{\mathcal{Y}}|_{\mathcal{V}}}, \; \text{GCA recall} = \frac{|\mathcal{S}^{\mathcal{X}}|_{\mathcal{V}}}{|\mathcal{G}^{\mathcal{X}}|_{\mathcal{V}}},$$

In above, $\mathcal{S}^{\mathcal{Y}}$ and $\mathcal{G}^{\mathcal{Y}}$ represent a subset ($\mathcal{V} = \mathcal{Y}, \mathcal{E}$) of each graph $\mathcal{S}$ and $\mathcal{G}$, respectively. One of the drawbacks is that the new hyper-parameters $\epsilon$, $\eta_c$, and $\eta_q$ need to be set arbitrarily according to the image dataset in order to evaluate fake images well. In addition, even if we use the hyper-parameters that seem most appropriate, it is difficult to verify that the actual evaluation results are correct because they do not approximate the underlying feature distribution.

**Manifold Topology Divergence** (MTD). Barannikov et al. [28] proposes a new metric that uses the life-length of the $k$-dimensional homology of connected components between real features and fake features formed through Vietoris-Rips filtration [41]. MTD is simply defined as the sum of the life-length of homologies and is characterized by an evaluation trend consistent with the FID. Specifically, MTD constructs the graph $\mathcal{G}$ by connecting edges between features with a distance smaller than the threshold $\epsilon$. Then the birth and death of the $k$-dimensional homologies in $\mathcal{G}$ are recorded by adjusting the threshold $\epsilon$ from 0 to $\infty$. The life-length $l_i(h)$ is obtained through $death_i - birth_i$ of any $k$-dimensional homology $h$, and let life-length set of all homologies as $L(h)$. MTD repeats the process of obtaining $L(h)$ on randomly sampled subsets $\mathcal{X}' \in \mathcal{X}$ and $\mathcal{Y}' \in \mathcal{Y}$ at each iteration, and defined as the following:

$$MTD(\mathcal{X}, \mathcal{Y}) = \frac{1}{n} \sum_{i}^{n} L_i(h), \text{ where } L_i(h) \text{ is the life-length set defined at } i\text{th iteration.}$$

However, MTD has a limitation of only considering a fixed dimensional homology for evaluation. In addition, MTD lacks interpretability as it evaluates the model with a single assessment score.

# G   Philosophy of our metric & Practical Scenarios

## G.1   Philosophy of our metric

All evaluation metrics have different resolutions and properties. The philosophy of our metric is to propose a reliable evaluation metric based on statistically and topologically certain things. In a real situation, the data often contain outliers or noise, and these data points come from a variety of sources (e.g. human error, feature embedding network). These errors may play as outliers, which leads to an overestimation of the data distribution, which in turn leads to a false impression of improvement when developing generative models. As discussed in Section 2, P&R and its variants have different ways to estimate the supports, which overlook the possible presence and effect of noise. Naeem et al. [14] revealed that previous support estimation approaches may inaccurately estimate the true support under noisy conditions, and partially solved this problem by proposing to only use the real support. However, this change goes beyond the natural definition of precision and recall, and results in losing some beneficial properties like boundedness. Moreover, this is still a temporary solution since real data can also contain outliers. We propose a solution to the existing problem by minimizing the effects of noise, using topologically significant data points and retaining the natural definition of precision and recall.

## G.2   Practical Scenarios

From this perspective, we present two examples of realistic situations where outliers exist in the data and filtering out them can have a significant impact on proper model analysis and evaluation. With real data, there are many cases where outliers are introduced into the data due to human error [17, 18]. Taking the simplest MNIST as an example, suppose our task of interest is to generate 4. Since image number 7 is included in data set number 4 due to incorrect labeling (see Figure 1 of [17]), the support of the real data in the feature space can be overestimated by such outliers, leading to an unfair evaluation of generative models (as in Section 5.1.2 and 5.2.2); That is, the sample generated with weird noise may be in the overestimated support, and existing metrics without taking into account the reliability of the support could not penalize this, giving a good score to a poorly performing generator.

A similar but different example is when noise or distortions in the captured data (unfortunately) behave adversely on the feature embedding network used by the current evaluation metrics (as in Section 5.1.2 and 5.2.2); e.g., visually it is the number 7, but it is mismapped near the feature space where there are usually 4 and becomes an outlier. Then the same problem as above may occur. Note that in these simple cases, where the definition of outliers is obvious with enough data, one could easily examine the data and exclude outliers a priori to train a generative model. In the case of more complicated problems such as the medical field [18], however, it is often not clear how outliers are to be defined. Moreover, because data are often scarce, even outliers are very useful and valuable in practice for training models and extracting features, making it difficult to filter outliers in advance and decide not to use them.

On the other hand, we also provide an example where it is very important to filter out outliers in the generator sample and then evaluate them. To evaluate the generator, samples are generated by sampling from the preset latent space (typically Gaussian). Even after training is complete and the generators' outputs are generally fine, there's a latent area where generators aren't fully trained. Note that latent space sampling may contain samples from regions that the generator does not cover well during training ("unfortunate outlier"). When unfortunate outliers are included, the existing evaluation metrics may underestimate or overestimate the generator's performance than its general performance. (To get around this, it is necessary to try this evaluation several times to statistically stabilize it, but this requires a lot of computation and becomes impractical, especially when the latent space dimension is high.)

Especially considering the evaluation scenario in the middle of training, the above situation is likely to occur due to frequent evaluation, which can interrupt training or lead to wrong conclusions. On the other hand, we can expect that our metric will be more robust against the above problem since it pays more attention to the core (samples that form topologically meaningful structures) generation performance of the model.

### G.3 Details of the noise framework in the experiments

We have assessed the robustness of TopP&R against two types of non-IID noise perturbation through toy data experiments (in Section 5.1.3) and real data experiments (in Section 5.2.2). The scatter noise we employed consists of randomly extracted noise from a uniform distribution. This noise, even when extensively added to our data, does not form a data structure with any significant signal (topological feature). In other words, from the perspective of TopP&R, the formation of topologically significant data structures implies that data samples should possess meaningful probability values in the feature space. However, noise following a uniform distribution across all feature dimensions holds very small probability values, and thus, it does not constitute a topologically significant data structure. Consequently, such noise is filtered out by the bootstrap bands $c_{\mathcal{X}}$ or $c_{\mathcal{Y}}$ that we approximate (see Section 2).

The alternate noise we utilized in our experiments, swap noise, possesses distinct characteristics from scatter noise. Initially, given the real and fake data that constitute important data structures, we introduce swap noise by randomly selecting real and fake samples and exchanging their positions. In this process, the swapped fake samples follow the distribution of real data, and conversely, the real samples adhere to the distribution of fake data. This implies the addition of noise that follows the actual data distribution, thereby generating significant data structures. When the number of samples undergoing such positional swaps remains small, meaningful probability values cannot be established within the distribution. Statistically, an extremely small number of samples cannot form a probability distribution itself. As a result, our metric operates robustly in the presence of noise. However, as the count of these samples increases and statistically significant data structures emerge, TopP&R estimates the precise support of such noise as part of their distribution. By examining Figure 4 and 5, as well as Table A12, it becomes evident that conventional metrics struggle with accurate support estimation due to vulnerability to noise. This limitation results in an inability to appropriately evaluate aspects involving minority sets or data forming long distributions.

### G.4 Limitations

Since the KDE filtration requires extensive computations in high dimensions, a random projection that preserves high-dimensional distance and topological properties is inevitable. Based on the topological structure of the features present in the embedded low dimensional space, we have shown that TopP&R theoretically and experimentally has several good properties such as robustness to the noise from various sources and sensitivity to small changes in distribution. However, matching the distortion from the random projection to the noise level allowed by the confidence band is practically infeasible: The bootstrap confidence band is of order $\left(nh_n^d\right)^{-\frac{1}{2}}$, and hence for the distortion from the random projection to match this confidence band, the embedded dimension $d$ should be of order $\Omega\left(nh_n^d \log n\right)$ from Remark (B.2). However, computing the KDE filtration in dimension $\Omega\left(nh_n^d \log n\right)$ is practically infeasible. Hence in practice, the topological distortion from the random projection is not guaranteed to be filtered out by the confidence band, and there is a possibility of obtaining a less accurate evaluation score compared to calculations in the original dimensions.

Exploring the avenue of localizing uncertainty at individual data points separately presents an intriguing direction of research. Such an approach could potentially be more sample-efficient and disregard less data points. However, considering our emphasis on preserving topological signals, achieving localized uncertainty estimation in this manner is challenging within the current state of the art. For instance, localizing uncertainty in the kernel density estimate $\hat{p}$ is feasible, as functional variability is inherently local. To control uncertainty at a specific point $x$, we primarily need to analyze the function value $\hat{p}(x)$ at that point. In contrast, localizing uncertainty for homological features situated at point $x$ requires the analysis of all points connected by the homological feature. Consequently, even if our intention is to confine uncertainty to a local point, it demands estimating the uncertainty at the global structural level. This makes localizing the uncertainty for topological features difficult given the tools in topology and statistics we currently have.

# H  Experimental Details

## H.1  Implementation details of embedding

We summarize the detailed information of our embedding networks implemented for the experiments. In Figure 2, 3, 4, A10, A4, and 5, `P&R` and `D&C` are computed from the features of ImageNet pre-trained VGG16 (fc2 layer), and `TopP&R` is computed from features placed in $\mathbb{R}^{32}$ with additional random linear projection. In the experiment in Figure A10, the SwAV embedder is additionally considered. We implement ImageNet pre-trained InceptionV3 (fc layer), VGG16 (fc2 layer), and SwAV as embedding networks with random linear projection to 32 dimensional feature space to compare the ranking of GANs in Table 1.

## H.2  Implementation details of confidence band estimator

---
**Algorithm 1** Confidence Band Estimator

---
\# KDE: kernel density estimator
\# h: kernel bandwidth parameter
\# k: number of repeats
\# $\hat{\theta}$: set of difference

Given $\mathcal{X} = \{X_1, X_2, \ldots, X_n\}$
$\hat{p}_h = KDE(\mathcal{X})$
**for** $iteration = 1, 2, \ldots, k$ **do**
   # Define $\mathcal{X}^*$ with bootstrap sampling
   $\mathcal{X}^* =$ random sample $n$ times with repeat from $\mathcal{X}$
   # $\hat{p}_h^*$ replaces population density
   $\hat{p}_h^* = KDE(\mathcal{X}^*)$
   # compute $\hat{\theta}$ with bootstrap samples
   Append $\hat{\theta}$ with $\sqrt{n}||\hat{p}_h - \hat{p}_h^*||_\infty$
**end for**

\# grid search for the confidence band
**for** $q \in [min(\hat{\theta}), max(\hat{\theta})]$ **do**
   count = 0
   **for** $element \in \hat{\theta}$ **do**
      **if** $element > q$ **then**
         $count = count + 1$
      **end if**
   **end for**
   # define the band threshold
   **if** $count/k \approx \alpha$ **then**
      $q_\alpha = q$
   **end if**
**end for**
# define estimated confidence band
$c_\alpha = q_\alpha/\sqrt{n}$

---

## H.3  Choice of confidence level

For the confidence level $\alpha$, we would like to point out that $\alpha$ is not the usual hyperparameter to be tuned: It has a statistical interpretation of the probability or the level of confidence to allow error, noise, etc. The most popular choices are $\alpha = 0.1, 0.05, 0.01$, leading to 90%, 95%, 99% confidence. We used $\alpha = 0.1$ throughout our experiments.

## H.4  Estimation of Bandwidth parameter

As we discussed in Section 2, since `TopP&R` estimates the manifold through KDE with kernel bandwidth parameter $h$, we need to approximate it. The estimation techniques for $h$ are as follows: (**a**) a method of selecting $h$ that maximizes the survival time ($S(h)$) or the number of significant homological features ($N(h)$) based on information obtained about persistent homology using the filtration method, (**b**) a method using the median of the k-nearest neighboring distances between features obtained by the balloon estimator (for more details, please refer to [42], [43], and [44]). Note that, the bandwidths $h$ for all the experiments in this paper are estimated via Balloon Bandwidth Estimator.

For (**a**), following the notation in Section A, let the $i$th homological feature of persistent diagram be $(b_i, d_i)$, then we define its life length as $l_i(h) = d_i - b_i$ at kernel bandwidth $h$. With confidence band $c_\alpha(h)$, we select h that maximizes one of the following two quantities:

$$N(h) = \#\{i : l_i(h) > c_\alpha(h)\}, \ S(h) = \sum_i [l_i(h) - c_\alpha(h)]_+.$$

Note that, we denote the confidence band $c_\alpha$ as $c_\alpha(h)$ considering the kernel bandwidth parameter $h$ of KDE in Algorithm 1.

For (**b**), the balloon bandwidth estimator is defined as below:

**Algorithm 2** Balloon Bandwidth Estimator

| | |
|---|---|
| # $h$: kernel bandwidth | $d(X_{idx}, \mathcal{X}) = \|X_{idx} - \mathcal{X}\|_2^2,$ |
| # KND: kth nearest distance | i.e., $\{d(X_{idx}, X_1), d(X_{idx}, X_2), \ldots\}$ |
| # idx: index | # Calculate kth nearest neighbor distance |
| # sort: sort in ascending order | $KND_{idx} = \text{sort}(d(X_{idx}, \mathcal{X}))[k]$ |
| | **end for** |
| Given $\mathcal{X} = \{X_1, X_2, \ldots, X_n\}$ | Given $KND = \{KND_1, KND_2, \ldots, KND_n\}$ |
| **for** $idx = 1, 2, \ldots, n$ **do** | # Define the estimated bandwidth $\hat{h}$ |
|   # Compute L2 distance between $X_{idx}$ and $\mathcal{X}$ | $\hat{h} = \text{median}(KND)$ |

## H.5 Computational complexity

TopP&R is computed using the confidence band $c_\alpha$ and KDE bandwidth $h$. The computational cost of the confidence band calculation (Algorithm 1) and the balloon bandwidth calculation (Algorithm 2) are $O(k * n^2 * d)$ and $O(n^2 * d)$, respectively, where $n$ represents the data size, $d$ denotes the data dimension, and $k$ stands for the number of repeats ($k = 10$ in our implementation). The resulting computational cost of TopP&R is $O(k * n^2 * d)$. In our experiments based on real data, calculating TopP&R once using real and fake features, each consisting of 10k samples in a 4096-dimensional space, takes approximately 3-4 minutes. This computation speed is notably comparable to that of P&R and D&C, utilizing CPU-based computations. Note that, P&R and D&C that we primarily compare in our experiments are algorithms based on $k$-nearest neighborhood for estimating data distribution, and due to the computation of pairwise distances, the computational cost of these algorithm is approximately $O(n^2 * d)$.

Table A1: Wall clock, CPU time and Time complexity for evaluation metrics for 10k real and 10k fake features in 4096 dimension. For TopP&R, $r$ is random projection dimension ($r \ll d$). The results are measured by the time module in python.

| Time | PR | DC | **TopPR (Ours)** |
|---|---|---|---|
| Wall Clock | 1min 43s | 1min 40s | 1min 58s |
| CPU time | 2min 44s | 2min 34s | 2min 21s |
| Time complexity | $O(n^2 * d)$ | $O(n^2 * d)$ | $O(k * n^2 * r)$ |

## H.6 Mean hamming distance

Hamming distance (HD) [45] counts the number of items with different ranks between A and B, then measures how much proportion differs in the overall order, i.e. for $A_i \in A$ and $B_i \in B$, $HD(A, B) \coloneqq \sum_{i=1}^n 1\{k : A_i \neq B_i\}$ where k is the number of differently ordered elements. The **mean HD** is calculated as follows to measure the average distances of three ordered lists: Given three ordered lists $A, B,$ and $C$, $\bar{HD} = (HD(A, B) + HD(A, C) + HD(B, C))/3$.

## H.7 Explicit values of bandwidth parameter

Since our metric adaptively reacts to the given samples of $P$ and $Q$, we have two $h$'s per experiment. For example, in the translation experiment (Figure 2), there are 13 steps in total, and each time we estimate $h$ for $P$ and $Q$, resulting in a total of 26 $h$'s. To show them all at a glance, we have listed all values in one place. We will also provide the code that can reproduce the results in our experiments upon acceptance.

Table A2: Bandwidth parameters $h$ of distribution shift in Section 5.1.1.

| $\mu$ | $\leftarrow$ shift | | | | | | | | | | | | shift $\rightarrow$ |
|---|---|---|---|---|---|---|---|---|---|---|---|---|---|
| real h | 8.79 | 8.60 | 8.58 | 8.67 | 8.63 | 8.71 | 8.66 | 8.73 | 8.54 | 8.63 | 8.54 | 8.75 | 8.78 |
| fake h | 8.46 | 8.86 | 8.60 | 8.49 | 8.44 | 8.80 | 8.55 | 8.63 | 8.74 | 8.61 | 8.58 | 8.83 | 8.60 |

Table A3: Bandwidth parameters $h$ of sequential mode drop in Section 5.1.2.

| Steps | 0 | 1 | 2 | 3 | 4 | 5 | 6 |
|---|---|---|---|---|---|---|---|
| real h | 10.5 | 10.6 | 10.5 | 10.7 | 10.5 | 10.8 | 10.6 |
| fake h | 11.0 | 10.2 | 10.0 | 9.82 | 9.57 | 9.29 | 8.78 |

Table A4: Bandwidth parameters $h$ of simultaneous mode drop in Section 5.1.2.

| Steps | 0 | 1 | 2 | 3 | 4 | 5 | 6 | 7 | 8 | 9 | 10 |
|---|---|---|---|---|---|---|---|---|---|---|---|
| real h | 10.5 | 10.6 | 10.6 | 10.8 | 10.3 | 10.6 | 10.7 | 10.6 | 10.4 | 10.4 | 10.6 |
| fake h | 10.6 | 10.5 | 10.7 | 10.3 | 10.4 | 10.0 | 9.88 | 9.31 | 9.19 | 9.07 | 8.79 |

Table A5: Bandwidth parameters $h$ of non-IID noise perturbation in Section 5.1.3.

| Steps | | 0 | 1 | 2 | 3 | 4 | 5 | 6 | 7 | 8 | 9 |
|---|---|---|---|---|---|---|---|---|---|---|---|
| Scatter | real h | 8.76 | 8.71 | 8.78 | 8.59 | 8.85 | 8.71 | 8.61 | 8.97 | 9.00 | 8.95 |
| | fake h | 8.61 | 8.76 | 8.63 | 8.93 | 8.46 | 8.72 | 8.67 | 8.68 | 9.07 | 8.88 |
| Swap | real h | 8.65 | 8.44 | 8.76 | 8.62 | 8.55 | 8.75 | 8.96 | 8.59 | 8.53 | 8.74 |
| | fake h | 8.52 | 8.55 | 8.68 | 8.97 | 8.87 | 8.94 | 8.84 | 9.05 | 8.94 | 8.90 |

Table A6: Bandwidth parameters $h$ of FFHQ truncation trick in Section I.11.

| $\Psi$ | $\Psi \downarrow$ | | | | | | | | | | $\Psi \uparrow$ |
|---|---|---|---|---|---|---|---|---|---|---|---|
| real h | 6.26 | 6.41 | 6.31 | 6.23 | 6.06 | 6.24 | 6.20 | 6.61 | 6.39 | 6.28 | 6.26 |
| fake h | 1.04 | 1.67 | 2.36 | 2.90 | 3.41 | 3.93 | 4.31 | 4.90 | 5.51 | 5.60 | 6.46 |

Table A7: Bandwidth parameters $h$ of CIFAR10 sequential mode drop in Section 5.2.1.

| Steps | 0 | 1 | 2 | 3 | 4 | 5 | 6 | 7 | 8 | 9 |
|---|---|---|---|---|---|---|---|---|---|---|
| real h | 7.09 | 6.95 | 6.96 | 6.92 | 7.02 | 6.81 | 6.86 | 6.88 | 6.78 | 6.67 |
| fake h | 6.87 | 6.75 | 6.88 | 6.64 | 6.40 | 6.64 | 6.41 | 6.44 | 6.62 | 6.04 |

Table A8: Bandwidth parameters $h$ of CIFAR10 simultaneous mode drop in Section 5.2.1.

| Steps | 0 | 1 | 2 | 3 | 4 | 5 | 6 | 7 | 8 | 9 | 10 |
|---|---|---|---|---|---|---|---|---|---|---|---|
| real h | 6.78 | 6.95 | 7.05 | 6.62 | 6.85 | 6.51 | 6.94 | 6.91 | 6.80 | 7.00 | 6.78 |
| fake h | 6.75 | 6.69 | 7.14 | 6.68 | 6.75 | 6.84 | 6.61 | 6.68 | 6.50 | 6.32 | 6.17 |

Table A9: Bandwidth parameters $h$ of FFHQ non-IID noise perturbation in Section 5.2.2.

| Steps | | 0 | 1 | 2 | 3 | 4 | 5 | 6 | 7 | 8 | 9 |
|---|---|---|---|---|---|---|---|---|---|---|---|---|
| Scatter | real h | 7.24 | 7.03 | 7.01 | 7.45 | 7.38 | 7.66 | 7.46 | 7.36 | 7.79 | 7.66 |
| | fake h | 5.41 | 5.77 | 5.67 | 5.49 | 5.71 | 5.65 | 5.82 | 6.08 | 5.81 | 6.02 |
| Swap | real h | 7.25 | 7.20 | 6.99 | 6.67 | 7.07 | 6.96 | 7.03 | 6.94 | 6.83 | 6.86 |
| | fake h | 6.87 | 6.63 | 6.60 | 6.91 | 6.74 | 6.77 | 6.92 | 6.83 | 6.83 | 6.82 |

# I  Additional Experiments

## I.1  Survivability of minority sets in the long-tailed distribution

An important point to check in our proposed metric is the possibility that a small part of the total data (i.e., minority sets), but containing important information, can be ignored by the confidence band. We emphasize that since our metric takes topological features into account, even minority sets are not filtered conditioned that they have topologically significant structures. We assume that signals or data

Table A10: Bandwidth parameters $h$ of FFHQ noise addition in Section 5.2.3.

|  | Noise intensity | 0 | 1 | 2 | 3 |
|---|---|---|---|---|---|
| Gaussian noise | real h | 6.52 | 6.27 | 6.41 | 6.49 |
|  | fake h | 6.32 | 6.15 | 4.06 | 2.64 |
| Gaussian blur | real h | 6.08 | 6.26 | 6.26 | 6.47 |
|  | fake h | 6.12 | 5.13 | 4.60 | 3.40 |
| Black rectangle | real h | 6.06 | 6.28 | 6.49 | 6.44 |
|  | fake h | 6.36 | 6.66 | 6.31 | 5.93 |

Table A11: Bandwidth parameters $h$ of GAN ranking in Section 5.2.4.

|  |  | StyleGAN2 | ReACGAN | BigGAN | PDGAN | ACGAN | WGAN-GP |
|---|---|---|---|---|---|---|---|
| InceptionV3 | real h | 2.605 | 2.629 | 2.581 | 2.587 | 2.615 | 2.606 |
|  | fake h | 2.226 | 2.494 | 2.226 | 2.639 | 2.573 | 2.063 |
| VGG16 | real h | 7.732 | 8.021 | 8.175 | 7.792 | 7.845 | 8.036 |
|  | fake h | 7.880 | 6.839 | 7.178 | 6.174 | 6.839 | 3.615 |
| SwAV | real h | 0.774 | 0.765 | 0.785 | 0.780 | 0.780 | 0.822 |
|  | fake h | 0.740 | 0.667 | 0.6746 | 0.619 | 0.627 | 0.502 |

that are minority sets have topological structures, but outliers exist far apart and lack a topological structure in general.

To test this, we experimented with CIFAR10 [46], which has 5,000 samples per class. We simulate a dataset with the majority set of six classes (2,000 samples per class, 12,000 total) and the minority set of four classes (500 samples per class, 2,000 total), and an ideal generator that exactly mimics the full data distribution. As shown in the Table A12, the samples in the minority set remained after the filtering process, meaning that the samples were sufficient to form a significant structure. Both D&C and TopP&R successfully evaluate the distribution for the ideal generator. To check whether our metric reacts to the change in the distribution even with this harsh setting, we also carried out the mode decay experiment. We dropped the samples of the minority set from 500 to 100 per class, which can be interpreted as an 11.3% decrease in diversity relative to the full distribution (Given (a) ratio of the number of samples between majority and minority sets $= 12,000 : 2,000 = 6 : 1$ and (b) 80% decrease in samples per minority class, the true decay in the diversity is calculated as $\frac{1}{(1+6)} \times 0.8 = 11.3\%$ with respect to the enitre samples). Here, recall and coverage react somewhat less sensitively with their reduced diversities as 3 p.p. and 2 p.p., respectively, while TopR reacted most similarly (9 p.p.) to the ideal value. In summary, TopP&R shows much more sensitiveness to the changes in data distribution like mode decay. Thus, once the minority set has survived the filtering process, our metric is likely to be much more responsive than existing methods.

Table A12: Proportion of surviving minority samples in the long-tailed distribution after the noise exclusion with confidence band $c_{\mathcal{X}}$. The p.p. indicates the percentage points.

|  | Before mode drop | After mode drop |
|---|---|---|
| **Proportion of survival** (before / after filtering) | 100% $\to$ 59% | 100% $\to$ 57% |
| **TopP&R** (Fidelity / Diversity) | 0.99 / 0.96 | 0.99 / 0.87 (9 p.p $\downarrow$) |
| P&R (Fidelity / Diversity) | 0.73 / 0.73 | 0.74 / 0.70 (3 p.p $\downarrow$) |
| D&C (Fidelity / Diversity) | 0.99 / 0.96 | 1.02 / 0.94 (2 p.p $\downarrow$) |

## I.2 Experiment on Non-IID perturbation with outlier removal methods

We compared the performance of metrics on a denoised dataset using Local Outlier Factor (LOF) [47] and Isolation Forest (IF) [48]. The experimental setup is identical to that of Figure 4 (see Section G.3 for details of our noise framework). In this experiment, we applied the outlier removal method before calculating P&R and D&C. From the result, P&R and D&C still do not provide a stable evaluation, while TopP&R shows the most consistent evaluation for the two types of noise without changing its trend.

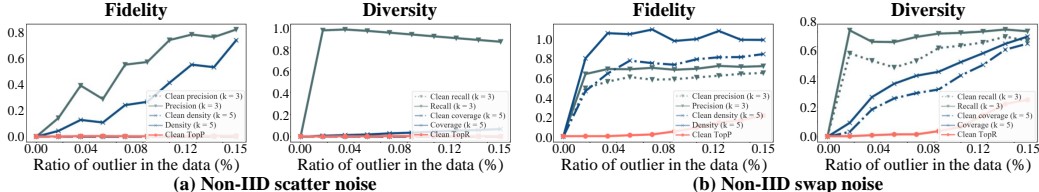

Figure A2: Behaviors of evaluation metrics on Non-IID perturbations. The dotted line in the graph shows the performance of metrics after the removal of outliers using the IF. We use "Clean" as a prefix to denote the evaluation after IF.

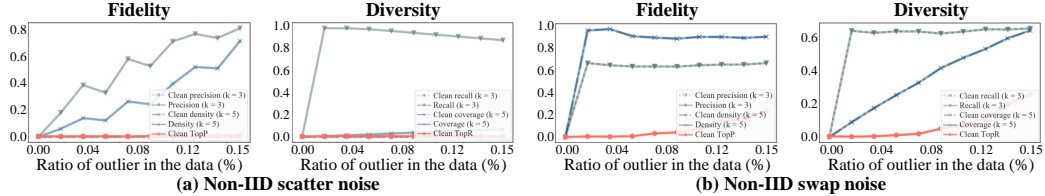

Figure A3: Behaviors of evaluation metrics on Non-IID perturbations. The dotted line in the graph shows the performance of metrics after the removal of outliers using the LOF. We use "Clean" as a prefix to denote the evaluation after LOF.

Since `TopP&R` can discriminate topologically significant signals by estimating the confidence band (using KDE filtration function), there is no need to arbitrarily set cutoff thresholds for each dataset. On the other hand, in order to distinguish inliers from specific datasets using LOF and IF methods, a threshold specific to the dataset must be arbitrarily set each time, so there is a clear constraint that different results are expected each time for each parameter setting (which is set by the user). Through this, it is confirmed that our evaluation metric guarantees the most consistent scoring based on the topologically significant signals. Since the removal of outliers in the `TopP&R` is part of the support estimation process, it is not appropriate to replace our unified process with the LOF or IF. In detail, `TopP&R` removes outliers through a clear threshold called confidence band which is defined by KDE, and this process simultaneously defines KDE's super-level set as the estimated support. If this process is separated, the topological properties and interpretations of estimated support also disappear. Therefore, it is not practical to remove outliers by other methods when calculating `TopP&R`.

### I.3 Sequential and simultaneous mode dropping with real dataset

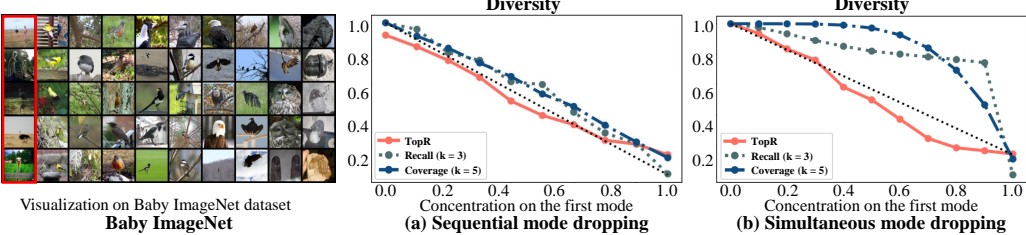

Figure A4: Comparison of evaluation metrics under sequential and simultaneous mode dropping scenario with Baby ImageNet[15].

### I.4 Sensitiveness to noise intensity

The purpose of this experiment (Figure A5) is to closely observe the noise sensitivity of metrics in Figure 6. As the noise intensity is incrementally increased until all the metrics converge to 0, the most ideal outcome for the metrics is to exhibit a linear trend to capture these changes most effectively. From the results, it is able to observe that both `TopP&R` and `P&R` exhibit the most linear trend in their evaluation outcomes, while `D&C` demonstrates that least capability to reflect differences based on noise intensity.

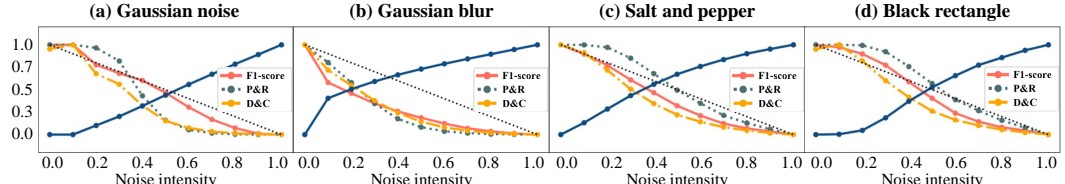

Figure A5: Verification of whether `TopP&R` can make an accurate quantitative assessment of noise image features. Gaussian noise, gaussian blur, salt and pepper, and black rectangle noise are added to the FFHQ images and emedded with T4096. The noise intensity is linearly increased until all metrics converge to zero.

## I.5 Evaluating state-of-the-art generative models on the ImageNet

We conducted our evaluation using the ImageNet dataset, which covers a wide range of classes. We employed commonly accepted metrics in the community, including FID and KID, to rank the considered generative models effectively. The purpose of this experiment is not only to establish the applicability of our ranking metric for generative models, akin to existing single-score metrics, but also to emphasize its interpretability. In Table A13, we calculated the HD between `P&R` variants and the more reliable KID metric. This comparison aimed to assess the alignment of our metric with established single-score metrics. The results consistently showed that `TopP&R`'s evaluation results are the most consistent with single-score metrics across various embedding networks. Conversely, `P&R` exhibited an inconsistent evaluation trend compared to KID, which struggled to effectively differentiate between ReACGAN and BigGAN. Note that, Kang et al. [15] previously demonstrated that BigGAN performs worse than ReACGAN. However, surprisingly, `P&R` consistently rated BigGAN as better than ReACGAN across all embedding networks. These findings imply that the `P&R`'s unreliable support estimation under noisy conditions might hinder its ability to accurately distinguish between generative models.

Table A13: Ranking results based on FID, KID, MTD, `TopP&R`, `P&R` and `D&C` for ImageNet (128 × 128) trained generative models. SwAV, VGG16, and InceptionV3 represent the embedders, and the numbers in parentheses indicate the ranks assigned by each metric to the evaluated models. The HD is the hamming distance between KID and metrics.

| | Model | ADM | StyleGAN-XL | ReACGAN | BigGAN | HD |
|---|---|---|---|---|---|---|
| **InceptionV3** | **FID** ($\downarrow$) | 9.8715 (1) | 10.943 (2) | 24.409 (3) | 33.935 (4) | 0 |
| | **KID** ($\downarrow$) | 0.0010 (1) | 0.0025 (2) | 0.0062 (3) | 0.0205 (4) | - |
| | **TopP&R** ($\uparrow$) | 0.9168 (1) | 0.8919 (2) | 0.8886 (3) | 0.7757 (4) | 0 |
| | D&C ($\uparrow$) | 1.0333 (2) | 1.0587 (1) | 0.8468 (3) | 0.5400 (4) | 2 |
| | P&R ($\uparrow$) | 0.6988 (1) | 0.6713 (2) | 0.4462 (4) | 0.5079 (3) | 2 |
| | MTD ($\downarrow$) | 1.7308 (2) | 1.5518 (1) | 2.2122 (3) | 3.0891 (4) | 2 |
| **VGG16** | **TopP&R** ($\uparrow$) | 0.9778 (1) | 0.8896 (2) | 0.8376 (3) | 0.5599 (4) | 0 |
| | D&C ($\uparrow$) | 1.0022 (3) | 0.9159 (4) | 1.1322 (1) | 1.1020 (2) | 4 |
| | P&R ($\uparrow$) | 0.7724 (2) | 0.7785 (1) | 0.4092 (4) | 0.5237 (3) | 4 |
| | MTD ($\downarrow$) | 15.748 (3) | 22.313 (4) | 10.268 (1) | 13.992 (2) | 4 |
| **SwAV** | **TopP&R** ($\uparrow$) | 0.8851 (2) | 0.9102 (1) | 0.6457 (3) | 0.4698 (4) | 2 |
| | D&C ($\uparrow$) | 1.0717 (1) | 0.8898 (4) | 1.0654 (2) | 1.0578 (3) | 3 |
| | P&R ($\uparrow$) | 0.6096 (1) | 0.5596 (2) | 0.1335 (4) | 0.1544 (3) | 2 |
| | MTD ($\downarrow$) | 0.4165 (3) | 0.7789 (4) | 0.3601 (2) | 0.3200 (1) | 4 |

Table A14: Ranking results based on `TopP&R`, `P&R`, and `D&C` for ImageNet (128 × 128) trained generative models. SwAV, VGG16, and InceptionV3 represent the embedders, and the numbers in parentheses indicate the ranks assigned by each metric to the evaluated models. Note that, the random projection is applied to `TopP&R`, `P&R`, and `D&C`.

| | Model | ADM | StyleGAN-XL | ReACGAN | BigGAN |
|---|---|---|---|---|---|
| **InceptionV3** | **TopP&R** (↑) | 0.9168 (1) | 0.8919 (2) | 0.8886 (3) | 0.7757 (4) |
| | D&C w/ rand proj (↑) | 1.0221 (2) | 1.0249 (1) | 0.9218 (3) | 0.7099 (4) |
| | P&R w/ rand proj (↑) | 0.7371 (1) | 0.7346 (2) | 0.6117 (4) | 0.6305 (3) |
| **VGG16** | **TopP&R** (↑) | 0.9778 (1) | 0.8896 (2) | 0.8376 (3) | 0.5599 (4) |
| | D&C w/ rand proj (↑) | 1.0086 (3) | 0.9187 (4) | 1.0895 (1) | 1.0803 (2) |
| | P&R w/ rand proj (↑) | 0.7995 (2) | 0.8056 (1) | 0.5917 (4) | 0.6304 (3) |
| **SwAV** | **TopP&R** (↑) | 0.8851 (2) | 0.9102 (1) | 0.6457 (3) | 0.4698 (4) |
| | D&C w/ rand proj (↑) | 1.0526 (3) | 0.9479 (4) | 1.1037 (2) | 1.1127 (1) |
| | P&R w/ rand proj (↑) | 0.6873 (1) | 0.6818 (2) | 0.4179 (3) | 0.4104 (4) |

## I.6 Verification of random projection effect in generative model ranking

We also applied the same random projection used by `TopP&R` to `P&R` and `D&C` in Section I.6. From the results of the Table A14, it is evident that the application of random projection to `P&R` leads to different rankings than its original evaluation tendencies. Similarly, the rankings of `D&C` with random projection still exhibit significant variations based on the embedding. Thus, we have demonstrated that random projection does not provide consistent evaluation tendencies across different embeddings, while also highlighting that this characteristic is unique to `TopP&R`. To quantify these differences among metrics, we computed the mean Hamming Distance (MHD) between the results in the embedding spaces of each metric. The calculated MHD scores were 1.33 for `TopP&R`, 2.67 for `P&R`, and 3.33 for `D&C`, respectively.

## I.7 Verifying the effect of random projection to the noisy data

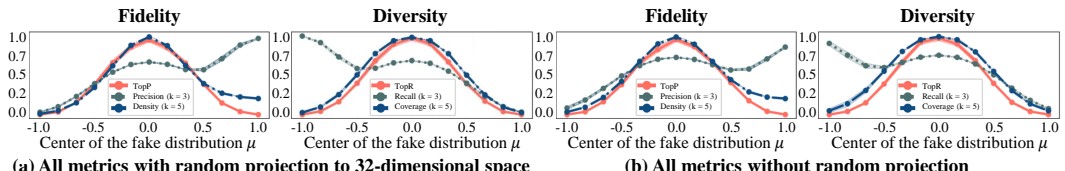

(a) All metrics with random projection to 32-dimensional space    (b) All metrics without random projection

Figure A6: Behaviors of evaluation metrics for outliers on real and fake distribution. For both real and fake data, the outliers are fixed at $3 \in \mathbb{R}^d$ where d is a dimension, and the parameter $\mu$ is shifted from -1 to 1. (a) depicts the comparison of all metrics using a random projection from 64 dimensions to 32 dimensions. (b) illustrates the difference between metrics without using random projection, where real and fake data are considered in 32 dimensions.

We experiment to ascertain whether utilizing a random projection address the drawbacks of existing metrics that are susceptible to noise, and we also conduct tests to validate whether `TopP&R` possesses noise robustness without using random projection. Through Figure A6 (a), it is evident that `P&R` and `D&C` still retain vulnerability to noisy features, and it is apparent that random projection itself does not diminish the impact of noise. Furthermore, Figure A6 (b) reveals that `TopP&R`, even without utilizing a random projection, exhibits robustness to noise through approximating the data support based on statistically and topologically significant features.

Building upon the insight from the toy data experiment in Figure A6 that random projection does not confer noise robustness to `P&R` and `D&C`, we aim to further demonstrate this fact through a real data experiment. In the experiment depicted in Figure A7, we follow the same setup as the previous experiment in Figure 5, applying the same random projection to `P&R` and `D&C` only. The results of

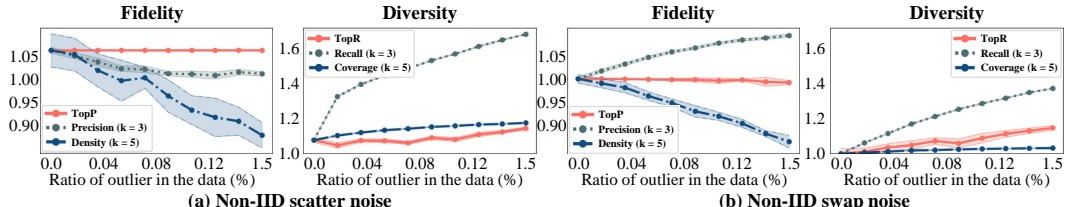

Figure A7: Comparison of evaluation metrics on Non-IID perturbations using FFHQ dataset. We replaced certain ratio of $\mathcal{X}$ and $\mathcal{Y}$ (a) with outliers and (b) by switching some of real and fake features. `P&R` and `D&C` were computed using the same 32-dimensional random projection.

this experiment affirm that random projection does not provide additional robustness against noise for `P&R` and `D&C`, while `TopP&R` continues to exhibit the most robust performance.

## I.8   Robustness of `TopP&R` with respect to random projection dimension

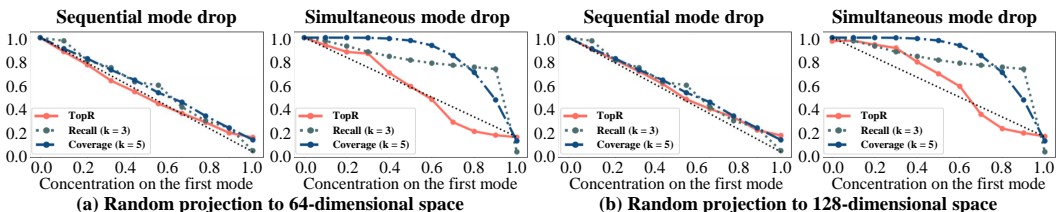

Figure A8: Comparison of evaluation trends of `TopP&R` with respect to random projection dimension in both sequential and simultaneous mode drop scenarios. Baby ImageNet dataset is used for the comparison, and `TopP&R` is computed using random projections to (a) 64 dimensions and (b) 128 dimensions.

To investigate whether the dimension of random projection influences the evaluation trend of `TopP&R`, we conduct mode dropping experiment using the Baby ImageNet dataset, similar to the experiment in Section I.3. Results in (a) and (b) of Figure A8 respectively compare the performance of `TopP&R` using random projection in dimensions of 64 and 128 with other metrics. From the experimental results, we observe that `TopP&R` is the most sensitive to the distributional changes, akin to the tendencies observed in the previous toy mode dropping experiment. Furthermore, upon examining the differences across random projection dimensions, `TopP&R` shows consistent evaluation trends regardless of dimensions.

## I.9   Trucation trick

$\psi$ is a parameter for the truncation trick and is first introduced in [4] and [1]. We followed the approach in [4] and [1]. GANs generate images using the noise input $z$, which follows the standard normal distribution $\mathcal{N}(0, I)$ or uniform distribution $\mathcal{U}(-1, 1)$. Suppose GAN inadvertently samples noise outside of distribution, then it is less likely to sample the image from the high density area of the image distribution $p(z)$ defined in the latent space of GAN, which leads to generate an image with artifacts. The truncation trick takes this into account and uses the following truncated distribution. Let $f$ be the mapping from the input to the latent space. Let $w = f(z)$, and $\bar{w} = \mathbb{E}[f(z)]$, where $z$ is either from $\mathcal{N}(0, I)$ or $\mathcal{U}(-1, 1)$. Then we use $w' = \bar{w} + \psi(w - \bar{w})$ as a truncated latent vector. If the value of $\psi$ increases, then the degree of truncation decreases which makes images have greater diversity but possibly lower fidelity.

## I.10   Toy experiment of trade-off between fidelity and diversity

We have designed a new toy experiment to mimic the truncation trick. With data distributions of 10k samples, $\mathcal{X} \sim \mathcal{N}(\mu = 0, I) \in \mathbb{R}^{32}$ and $\mathcal{Y} \sim \mathcal{N}(\mu = 0.6, \sigma^2) \in \mathbb{R}^{32}$, we measure the fidelity and diversity while incrementally increasing $\sigma$ from 0.7 to 1.3. For smaller $\sigma$ values, the evaluation metric should exhibit higher fidelity and lower diversity, while increasing $\sigma$ should demonstrate decreasing

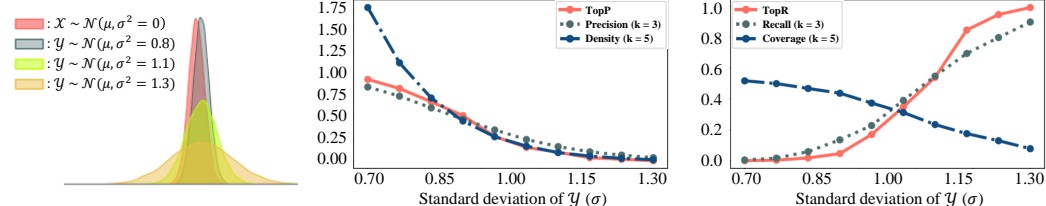

Figure A9: Trade-off between fidelity and diversity in a toy experiment. In this experiment, the $\mu$ of $\mathcal{X}$ was set to 0, and the $\mu$ of $\mathcal{Y}$ was set to 0.6. By incrementally increasing $\sigma$ from 0.7 to 1.3, the evaluation trend based on changes in the $\mathcal{Y}$ distribution is measured.

fidelity and higher diversity. From the results in Figure A9, both `TopP&R` and `P&R` exhibit a trade-off between fidelity and diversity.

## I.11 Resolving fidelity and diversity

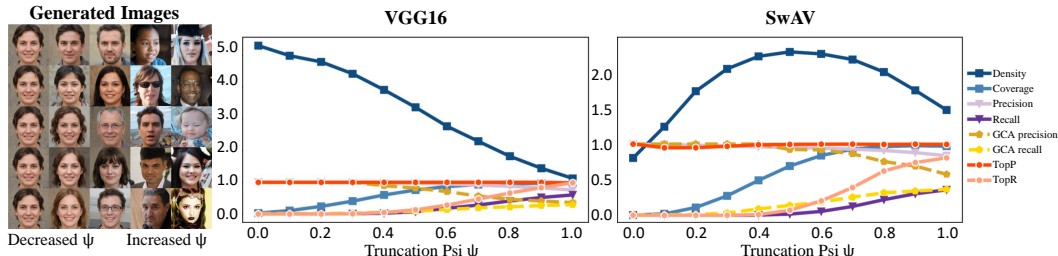

Figure A10: Behaviors of metrics with truncation trick. The horizontal axis corresponds to the value of $\psi$ denoting the increased diversity. The images are generated via StyleGAN2 with FFHQ dataset [1].

To test whether `TopP&R` responds appropriately to the change in the underlying distributions in real scenarios, we test the metric on the generated images of StyleGAN2 [2] using the truncation trick [1]. StyleGAN2, trained on FFHQ as shown by Han et al. [49], tends to generate samples mainly belonging to the majority of the real data distribution, struggling to produce rare samples effectively. In other words, while StyleGAN2 excels in generating high-fidelity images, it lacks diversity. Therefore, the evaluation trend in this experiment should reflect a steady high fidelity score while gradually increasing diversity. Han et al. [49] demonstrates that the generated image quality of StyleGAN2 that lies within the support region exhibits sufficiently realistic visual quality. Therefore, particularly in the case of `TopP&R`, which evaluates fidelity excluding noise, the fidelity value should remain close to 1. As shown in Figure A10, every time the distribution is transformed by $\psi$, `TopP&R` responds well and shows consistent behavior across different embedders with bounded scores in $[0, 1]$, which are important virtues as an evaluation metric. On the other hand, `Density` gives unbounded scores (fidelity > 1) and shows inconsistent trend depending on the embedder. Because `Density` is not capped in value, it is difficult to interpret the score and know exactly which value denotes the best performance (*e.g.*, in our case, the best performance is when `TopP&R` = 1).

