# Appendix

## A  More Background on Topological Data Analysis

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

$$\liminf_{r \to 0} \frac{P\left(\mathcal{B}(x, r)\right)}{r^d} > 0, \qquad \liminf_{r \to 0} \frac{Q\left(\mathcal{B}(y, r)\right)}{r^d} > 0.$$

*Remark* D.1. Assumption A1 is analogous to Assumption 2 of Kim et al. [37], but is weaker since the condition is pointwise on each $x \in \mathbb{R}^d$. And this condition is much weaker than assuming a density on $\mathbb{R}^d$: for example, a distribution supported on a low-dimensional manifold satisfies Assumption A1. This provides a framework suitable for high dimensional data, since many times high dimensional data lies on a low dimensional structure hence its density on $\mathbb{R}^d$ cannot exist. See Kim et al. [37] for a more detailed discussion.

For kernel functions, we assume the following regularity conditions:

**Assumption A2.** *Let $K : \mathbb{R}^d \to \mathbb{R}$ be a nonnegative function with $\left\|K\right\|_1 = 1$, $\left\|K\right\|_\infty, \left\|K\right\|_2 < \infty$, and satisfy the following:*

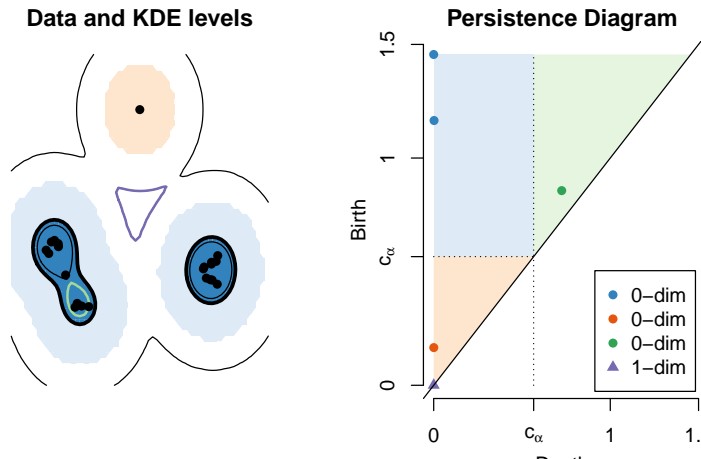

**Data and KDE levels**

**Persistence Diagram**

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

| | **KID** (↓) | 0.0010 (1) | 0.0025 (2) | 0.0062 (3) | 0.0205 (4) | - |
| | **TopP&R** (↑) | 0.9168 (1) | 0.8919 (2) | 0.8886 (3) | 0.7757 (4) | 0 |
| | D&C (↑) | 1.0333 (2) | 1.0587 (1) | 0.8468 (3) | 0.5400 (4) | 2 |
| | P&R (↑) | 0.6988 (1) | 0.6713 (2) | 0.4462 (4) | 0.5079 (3) | 2 |
| | MTD (↓) | 1.7308 (2) | 1.5518 (1) | 2.2122 (3) | 3.0891 (4) | 2 |
| **VGG16** | **TopP&R** (↑) | 0.9778 (1) | 0.8896 (2) | 0.8376 (3) | 0.5599 (4) | 0 |
| | D&C (↑) | 1.0022 (3) | 0.9159 (4) | 1.1322 (1) | 1.1020 (2) | 4 |
| | P&R (↑) | 0.7724 (2) | 0.7785 (1) | 0.4092 (4) | 0.5237 (3) | 4 |
| | MTD (↓) | 15.748 (3) | 22.313 (4) | 10.268 (1) | 13.992 (2) | 4 |
| **SwAV** | **TopP&R** (↑) | 0.8851 (2) | 0.9102 (1) | 0.6457 (3) | 0.4698 (4) | 2 |
| | D&C (↑) | 1.0717 (1) | 0.8898 (4) | 1.0654 (2) | 1.0578 (3) | 3 |
| | P&R (↑) | 0.6096 (1) | 0.5596 (2) | 0.1335 (4) | 0.1544 (3) | 2 |
| | MTD (↓) | 0.4165 (3) | 0.7789 (4) | 0.3601 (2) | 0.3200 (1) | 4 |