# OpenReview forum: "TopP&R: Robust Support Estimation Approach for Evaluating Fidelity and Diversity in Generative Models"
_NeurIPS.cc/2023/Conference — NeurIPS 2023 poster_

### Official Review · Reviewer_4EiX · 2023-06-21

**Soundness:** 3 good
**Presentation:** 1 poor
**Contribution:** 2 fair
**Rating:** 5
**Confidence:** 4

**Summary:**

The paper observes that the existing metrics for evaluating fidelity and diversity of generative models can be unreliable when outliers or non-IID perturbations are present. Thus, the authors propose a new metric based on Kernel Density Estimation under topological conditions that is more robust in the earlier scenarios. The main contributions of this paper are the metric that is robust to outliers and theoretical guarantees for robustness.

**Strengths:**

The paper assesses a very important topic of evaluating generative models and proposes a metric that provides robust estimates of sample fidelity and diversity in several toy/synthetic examples. Additionally, bringing concepts from topological data analysis (TDA) to evaluation of generative models might open new future directions in metric research.

**Weaknesses:**

The paper is difficult to follow for readers that are not very familiar with TDA and it could be improved significantly by adding intuition and a visual introduction to the most important concepts TDA and precision and recall. Furthermore, the concept of persistent homology, persistence diagram and confidence band estimation are discussed on a general level in Sec. 2 but the discussion seems disconnected how these are related to the proposed metric – adding concrete and intuitive examples would make this section more approachable. Unfortunately, I believe that in the current state paper is not very accessible to a wider audience.

The novelty of the work is challenging to evaluate and potentially limited. In particular, the proposed metric seems to rely on the removal of outliers before evaluating the Top P&R, and the Top P&R part of the metric seems quite similar to previous work. Ideally, I would hope to see real world use cases with state-of-the-art models, where the metric provides more reliable evaluation (now Tab. 1 Top P&R yields very similar results to improved P&R).

**Questions:**

1. The authors claim that robustness is required in practical evaluations of models since these cases are “wild”, i.e., might corrupt/outlier samples or the distributions might be sparse without density. I am curious as to why the proposed metric seems to perform similarly as improved PR with real generative models in Tab. 1? Also, Tab. 1 does not tell which dataset these models were trained and evaluated on.
Since the proposed metric is composed of an outlier removal and the Top P&R evaluation, how much of the performance in toy/synthetic datasets can be explained by the outlier removal part? Would adding a similar outlier removal to existing improved PR and DC improve their performance in the toy experiments?

2. I think detecting outliers might prove to be difficult in practice, compared to the toy experiments presented in the paper. Real datasets do not ground-truth “outlier labels” and in some datasets, e.g. in medical imaging, the distribution has a very long tail as there might be more data from patients without a certain disease than with the disease. In this case, modeling the patients with the disease might be more interesting than modeling patients without the disease. However, when evaluating Top P&R the contributions of the patients with the diseases might be suppressed as they are deemed as outliers.

3. How does the dimensionality of the random projection affect the performance of the metric?

4. Is the bandwidth parameter $h$ different for each model in Tab. 1? If so, is it possible that some models obtain artificially better scores because their outlier removal is different than for the other models? Are Top P&R scores with different bandwidth parameters directly comparable?

**Limitations:**

The authors adequately addressed the limitations of their work.

---

> ### Author Rebuttal · Authors · 2023-08-09
>
> ## Reviewer 5
> **R5-1. Dataset used for Table 1:** (G1-2) CIFAR-10. We apologize for not mentioning the datasets. We revised the caption of Table 1 in the revised paper.
>
> **Reason why TopP&R shows similar performance to P&R in Table 1:** (G1-3, G1-4). TopP&R shows better performance than the others when we apply it to more large-scale datasets, such as ImageNet and Baby ImageNet.
>
> **Application of outlier removal to improved P&R and D&C:** We had already compared P&R and D&C with TopP&R by applying the outlier removal method (e.g., Local Outlier Factor or Isolation Forest) to these methods in Section I.2. The experiments were conducted using the same setup as the experiments depicted in Figure 4 of the main text. By examining the results in Figures A2 and A3, it becomes evident that applying the outlier removal method to P&R and D&C still does not make them robust to noise.
> &nbsp;
> **R5-2. How TopP&R behaves with minority sets (e.g., long-tailed distributions):** If we correctly understand the reviewer's question, it pertains to whether TopP&R can effectively capture differences when there are variations in a minority set of patient data samples within a larger patient dataset, where only a small number of samples exhibit a specific disease. In our **Section I.1** of Appendix, we have already conducted experiments comparing how various metrics, including TopP&R, respond when changes occur in the minority set. This comparison was carried out to assess how different metrics react to variations in such a scenario.
>
> In **Table A11 of Section I.1**, our utilized data assumes a situation where out of a total of 10 classes, six data classes are majority classes, containing a substantial number of samples, while the remaining four minority classes have relatively small data counts, resulting in a long-tailed data distribution. In this context, we conducted experiments by further reducing the number of samples within the minority set, aiming to observe how TopP&R and other metrics respond to even the slightest changes in this distribution. Contrary to the reviewer's anticipation, the experimental results revealed that, among the metrics, TopP&R is the most sensitive in reflecting evaluation values influenced by subtle differences in the distribution. As detailed in Section C in Appendix, due to the inherent nature of TopP&R, if the minority set possesses a topologically and statistically significant data structure, these data points are deemed crucial features in approximating support. Therefore, considering all minority sets as outliers when estimating TopP&R’s support is an incorrect notion.
> &nbsp;
> **R5-3. The impact of changing the dimension of random projection on TopP&R:** (G2-2).
> &nbsp;
> **R5-4. Comparison of TopP&R scores across different bandwidth parameters:** (G3) In addition to our responses regarding $c_\alpha$ and $h$ in the general comments, $h$ being too small leads to the KDE with unstable topological features. On the other hand, $h$ being too large leads to the KDE with its topological features dissolved, so the choice of $h$ cannot be arbitrary but should be proper.
>
> **Whether certain models can achieve better performance due to different settings of the bandwidth parameter in Table 1:** (G1-2, G1-3) As stated in the response to the general comments, we have demonstrated that TopP&R provides a generally consistent ranking with metrics such as FID and KID, which are commonly used to assess performance differences among models in the field of generative modeling.
> &nbsp;
> **R5-W1. The paper is not easily understandable for individuals unfamiliar with TDA (+ more discussion between confidence band and proposed metric):** We can relate to the feedback provided by the reviewers, as this aspect was a significant concern during the paper writing process. Considering the limited page and the nature of the conference to which we submitted the paper, our approach was to structure the paper in a way that enables readers to understand the robustness of TopP&R at a high-level without getting entangled in intricate details of TDA. However, it is important to note that some aspects of TDA have been intentionally condensed in the main paper. For readers seeking a more comprehensive understanding of our work, we have provided the detailed explanation of TDA in the appendix.
>
> Furthermore, in response to the concerns raised by the reviewer, we will strive to provide more detailed explanations and visual introductions to address the aspects that require additional intuition. Our aim is to enhance the paper by offering a more comprehensive and visually informative presentation.
> &nbsp;
> **R5-W2. Novelty of this work:** Our proposed method introduces a novel theoretical approach by combining the traditional topic of consistency in statistical inference with topological data analysis. While existing literature has explored consistencies under i.i.d. settings without considering noise, and consistencies under noise frame framework, the focus on statistical inference in geometrical or topological settings with noise is relatively rare (e.g., Genovese, Christopher R., et al. (2012)). Our contribution lies in bridging this gap and providing a framework that incorporates both the consistency of topological analysis and the consideration of noise. This not only encompasses the theoretical properties of our method, called TopP&R, but also extends to identifying an appropriate noise framework and adapting statistical inference techniques for geometrical and topological data analysis.
> &emsp;&emsp;\- Genovese, Christopher R., et al. "Manifold estimation and singular deconvolution under Hausdorff loss." (2012): 941-963.
>
> **Real world use cases with SOTA model:** (G1-3).

---

> > ### Comment · Reviewer_4EiX · 2023-08-15
> >
> > Thank you for providing detailed answers to my feedback. I appreciate the authors efforts and acknowledge that examining and improving the evaluation metrics generative models is a significantly important topic. I am happy to update my score accordingly.

---

### Official Review · Reviewer_UDSM · 2023-07-06

**Soundness:** 3 good
**Presentation:** 3 good
**Contribution:** 3 good
**Rating:** 5
**Confidence:** 4

**Summary:**

The paper presents a novel evaluation metric called Topological Precision and Recall (TopP&R) for generative models. Existing metrics often suffer from unreliable support estimation and yield inconsistent results. In contrast, TopP&R systematically estimates supports by retaining topologically and statistically significant features with confidence. It provides robust evaluation, accurately capturing changes in samples even in the presence of outliers and non-independent and identically distributed perturbations. The metric combines ideas from Topological Data Analysis and statistical inference, offering a reliable approach to support estimation. The paper includes theoretical and experimental evidence to demonstrate the effectiveness and robustness of TopP&R, making it a valuable contribution to the evaluation of generative models.

**Strengths:**

The paper introduces a novel evaluation metric, Topological Precision and Recall (TopP&R), specifically designed for generative models. This metric addresses the limitations of existing evaluation metrics by focusing on robust support estimation, which is crucial for accurately assessing sample quality.

TopP&R provides a robust evaluation that is resilient to outliers and non-independent and identically distributed perturbations. The metric systematically estimates supports by retaining topologically and statistically significant features, ensuring reliable evaluation results even in challenging scenarios.

The paper demonstrates that TopP&R offers statistical consistency in its results. By incorporating concepts from Topological Data Analysis and statistical inference, the metric provides a systematic approach to distinguishing signal from noise and effectively capturing relevant features in the data.

The paper provides both theoretical and experimental evidence to support the effectiveness of TopP&R. The theoretical analysis establishes the robustness and consistency guarantees of the metric, while the experimental results demonstrate its superior performance compared to existing evaluation metrics.

TopP&R is a versatile evaluation metric that can be applied to various generative models. Its systematic approach to support estimation makes it suitable for evaluating the performance of models across different domains and datasets.

The authors provide code implementation for TopP&R, making it readily accessible for researchers and practitioners. This facilitates the adoption and reproducibility of the metric, contributing to the transparency and advancement of the field.

Overall, the paper's strengths lie in its introduction of a novel evaluation metric, its robustness and reliability in assessing sample quality, and its theoretical and experimental validation, demonstrating its effectiveness and broad applicability in the evaluation of generative models.

**Weaknesses:**

The paper mentions Algorithm 1 in line 107 but does not provide the algorithm or any further explanation in the paper. This lack of clarity leaves readers confused and makes it difficult to understand the specific steps and procedures involved.

Line 139 mentions the bootstrap bandwidth, but the paper does not clearly explain how this bandwidth is computed. It is important to provide details on the methodology used for computing the bandwidth to ensure transparency and reproducibility.

The paper lacks a clear discussion on the dimensionality of the problem that the proposed method can handle. It does not explain how the authors choose the dimensionality of the space for random projections or how they ensure that these projections capture relevant underlying features. Additionally, the paper does not provide theoretical results supporting the choices made in Section 3.4 regarding dimensionality and random projections.

In the experimental section, the authors choose a projection dimension of 32 without providing a clear justification or explanation for this specific choice. It is important to provide reasoning or empirical evidence to support the selection of this dimension and explain its impact on the performance of the method.

While the paper includes simulations to demonstrate the robustness and effectiveness of the proposed method, it lacks a thorough evaluation on real-world data where changes are not simulated. It would be valuable to see the performance of the method in real data scenarios to assess its practical applicability and validate its effectiveness in capturing genuine changes.

Addressing these weaknesses would strengthen the paper by providing clarity, offering detailed explanations for methodological choices, and conducting comprehensive evaluations on real-world data.

**Questions:**

No additional questions in addition to the weaknesses listed above.

**Limitations:**

Yes.

---

> ### Author Rebuttal · Authors · 2023-08-09
>
> ## Reviewer 4
> **R4-W1, W2. Explanation of line 107 and 139 (Algorithm 1):** Added. We apologize for missing the appropriate reference link to “Appendix H.2” for the algorithm. Unfortunately, this happened while we were breaking the manuscript into two (main and supplementary) documents for the submission.
> &nbsp;
> **R4-W3, W4 How the dimension of random projection was chosen:** (G2-2).
> &nbsp;
> **R4-W5. Evaluation on real-world data:** (G1).

---

> > ### Comment · Reviewer_UDSM · 2023-08-20
> >
> > After reading all the rebuttals, I have updated my score. There are a lot of choices on parameters made in the experimental setup without proper guidance which leaves practitioners wondering on the appropriate choice of such parameters. The authors should justify such arbitrary choices more rigorously.

---

> > > ### Author Response · Authors · 2023-08-20
> > >
> > > **There seems to be a misunderstanding regarding the hyper-parameter. In our algorithm, hyper-parameters are not arbitrarily set.** As mentioned in the official comment G3, the confidence level $\alpha$ is not a parameter for tuning, and furthermore, the kernel bandwidth $h$ that we employed in all of our experiments is automatically approximated through Algorithm 2 (Section H.4).
> > >
> > > Furthermore, **all other P&R variants employ a hyper-parameter denoted as $k$ (in $k$-means clustering or $k$-NN)**, which plays a similar role to $h$ of our algorithm. Moreover, existing metrics (except D&C) neither provide criteria nor algorithms for determining the appropriate $k$, often arbitrarily set, utilizing values of $k$ that have empirically shown some effectiveness on certain datasets. On the other hand, TopP&R is a more stable approach compared to conventional algorithms, as it considers meaningful topological and statistical characteristics of the data based on the confidence band (algorithm 1). This unique property allows our metric to provide robust and reliable evaluation criteria.

---

### Official Review · Reviewer_fxDp · 2023-07-08

**Soundness:** 3 good
**Presentation:** 3 good
**Contribution:** 2 fair
**Rating:** 4
**Confidence:** 3

**Summary:**

The paper introduces TopP&R, a comprehensive evaluation metric for generative models that enhances the accuracy of sample quality assessment in comparison to existing metrics. TopP&R incorporates topological data analysis and statistical inference, effectively estimating supports through the application of Kernel Density Estimator (KDE) under topological conditions. This metric demonstrates remarkable robustness towards outliers, successfully detects changes in distributions, and provides consistency guarantees and robustness for high-dimensional data with minimal assumptions. Notably, TopP&R is the pioneering metric that prioritizes robust support estimation, ensuring statistical consistency even when confronted with noise. The paper presents compelling theoretical findings and experimental evidence that strongly validate the effectiveness of TopP&R.

**Strengths:**

The paper introduces a novel evaluation metric, Topological Precision and Recall (TopP&R), for generative models. It combines topological data analysis and statistical inference to estimate support robustly and detect distributional changes accurately.

The paper provides theoretical and experimental evidence to demonstrate the reliability and consistency of the TopP&R metric. It accurately assesses sample quality, even in the presence of outliers, non-independent and identically distributed perturbations, and other noise conditions. The paper also presents a systematic approach to estimate support using Kernel Density Estimator (KDE) under topological conditions.

The paper clearly explains the motivation, limitations of existing metrics, and the design and implementation of the TopP&R metric. It includes visualizations and examples to enhance clarity and uses clear language to facilitate understanding.

The paper addresses the need for more reliable evaluation metrics for generative models. The proposed TopP&R metric fills this gap by providing a robust evaluation of sample quality and ensuring statistical consistency. Its findings have implications for improving the evaluation and development of generative models.

**Weaknesses:**

Limited accessibility: The paper may be challenging for readers unfamiliar with persistent homology and topological data analysis to grasp the methodology and results. Providing additional background information and explaining concepts in more detail would improve clarity.
Limited experimentation: The experiments are confined to a few datasets and models. Conducting experiments on a broader range of datasets and models would demonstrate the effectiveness of TopP&R in diverse scenarios.

Insufficient discussion of limitations: Although the authors mention some limitations, such as the requirement for a large number of samples to estimate the confidence band, a more comprehensive discussion of the limitations would help readers understand the scope and applicability of TopP&R.

Recommendations for improvement:

Critique existing metrics: Highlight limitations of commonly used metrics, such as Inception Score and Fréchet Inception Distance, due to unreliable support estimates from sample features.

Address noise vulnerability: Explain how TopP&R overcomes vulnerability to noise by considering topologically and statistically significant features with confidence.

Present experimental validation: Provide comprehensive experimental results comparing TopP&R to existing metrics, demonstrating its superior accuracy and consistency.

Discuss real-world applicability: Explore practical implications and applications of TopP&R in various domains, showcasing its relevance beyond theory.

Share open-source implementation: Make TopP&R's implementation publicly available, ideally on platforms like GitHub, to promote reproducibility, validation, and further advancements.

**Questions:**

1. How does TopP&R handle the trade-off between precision and recall in evaluating generative models?

2. Can you provide more details on the noise framework and statistical inference used in TopP&R?

3. Have you considered comparing TopP&R to other evaluation metrics based on topological features, such as Persistence Landscape?

4. Can you provide more information on the implementation of TopP&R and its applicability to different types of generative models?

5. How does the choice of threshold for retaining topologically and statistically significant features affect the performance of TopP&R?

6. Have you explored other techniques, like Gaussian mixture models or non-parametric methods, for estimating the density function in TopP&R?

**Limitations:**

Based on my understanding, the limitations of the TopP&R method mentioned in this paper can be summarized as follows:

1. The method requires a large number of samples to estimate the confidence band accurately, which can be a challenge for datasets with limited samples.

2. The reliance on the choice of kernel function and bandwidth can affect the performance of TopP&R, and the need for a theoretical guarantee on the choice of bandwidth can be difficult to obtain in practice.

3. The method assumes a smooth density function and independence between samples, which may not hold for all types of data, and the reliance on topological features may not capture all aspects of the data distribution.

---

> ### Author Rebuttal · Authors · 2023-08-09
>
> ## Reviewer 3
> **R3-1. Trade-off between precision and recall:** To quickly check whether TopP&R exhibits trade-off between fidelity and diversity as in improved P&R, we used $\mathcal{X}\sim{\mathcal{N}(0,1)}$ and $\mathcal{Y}\sim{\mathcal{N}(0.6, \sigma^2)}$ with a sample size of 10k and 32 dimensions. We compared the metrics while gradually increasing $\sigma$ from 0.7 to 1.3. The results of the experiment are shown in the table below.
> | $\sigma$     | 0.7  | 0.85 | 1.0  | 1.15 | 1.3  |
> |----|------|------|------|------|------|
> | Precision | 0.85 | 0.61 | 0.36 | 0.11 | 0.04 |
> | Recall    | 0.0  | 0.05 | 0.22 | 0.69 | 0.89 |
> | TopP      | 0.93 | 0.68 | 0.28 | 0.04 | 0.01 |
> | TopR      | 0.0  | 0.01 | 0.16 | 0.84 | 0.99 |
>
> &nbsp;
> **R3-2, W4. Provide more details on the noise framework and statistical inference:** Added. Noise refers to samples with low statistical probability values, not contributing significantly to the overall distribution. We discussed two non-IID noise types in Section 5.1.3 and 5.2.2: scatter noise and swap noise. Given the scatter noise is derived by adding noise sampled from a uniform distribution across all data dimensions, its influence remains limited due to its low probabilities. Consequently, evaluation metrics are expected to remain relatively stable even when the level of scatter noise increases. On the other hand, swap noise involves randomly selecting samples from two sets and exchanging their classes. At first, only a few swaps may not form a meaningful distribution, as the swapped samples don’t necessarily represent the distribution of the opposite set. However, as the level of swap noise increases, it can lead to the formation of significant distributions due to the heightened probability of class swaps.
>
> **How TopP&R overcomes vulnerability to noise:** The significant data structures possess meaningful probability values, while the opposite exhibits lower values. In this context, TopP&R approximate robust data support identifying significant data structures by the bootstrap band $c_\alpha$. In our paper, we have provided a detailed explanation of the noise framework we employed and described how our algorithm should effectively respond to such noise.
> &nbsp;
> **R3-3. Metrics utilizing topological features:** To the best of our knowledge, our paper compares all the metrics utilizing topological features  (e.g., GCA and MTD). We did not include a separate comparison with previous metrics such as Geometry Score (Khrulkov et al. 2018), as they are known to exhibit inferior performance compared to the modern topological feature-based metrics like MTD, and are also notoriously slow in computation, making them impractical for real-world usage.
>
> It is possible to define metrics using methods like the persistence landscape. For instance, one could construct landscapes using the filtration method for both real and fake datasets, and then define a new metric using statistical tests such as the t-test on the real and fake features. However, these methods utilizing persistence landscapes focus only on topological structures and forget distributional features not captured by persistence landscapes. As a result, **they may not effectively capture geometric transformations like translation in distributions**. Due to these characteristics, it is challenging to consider such methods particularly suitable as metrics for comparing generative models.
> &nbsp;
> **R3-4, W6, W7. Applicability to different types of generative models:** (G1-2, G1-3) TopP&R can be applied to all fields where generative models are used. For example, TopP&R can evaluate models that produce sound signals using appropriate networks to obtain features. Moreover, when comparing distribution similarity between various numeric data, TopP&R can be directly used even without an embedding network, treating each observation value as a feature.
>
> **Details of implementation, Code release:** (Table B2, Introduction) Please note that we have already provided the link to our code at the end of Section 1. Our metric primarily utilizes the Numpy, sklearn, and scipy libraries, and we utilize the linear layer provided by Pytorch. The metric computation is achievable through CPU operations. In terms of computation speed, our metric requires approximately 1 min 58 seconds of wall clock time to compare a real and fake dataset with a sample size of 10k. This indicates that our approach offers an appropriate calculation speed for effectively comparing generative models. Moreover, our algorithm is very user-friendly, as conducting experiments using our method only requires a simple one-line of code.
> &nbsp;
> **R3-5, L2. Choice of threshold:** (G3).
> &nbsp;
> **R3-6. Other techniques for estimating density:** (G4).
> &nbsp;
> **R3-W1. Limited experimentation:** (G1).
>
> **Addressing accessibility:** Given the constraints of limited page space and the conference’s nature, we designed the paper to allow readers to grasp the robustness of TopP&R at a high-level without becoming entangled in intricate TDA details. However, we want to emphasize that certain aspects of TDA were intentionally condensed in the main paper. For readers seeking a deeper understanding, we have provided a comprehensive TDA explanation in the appendix. We will make diligent efforts to furnish more detailed explanations to address areas that may require further clarity.
> &nbsp;
> **R3-W2, L1. Sample size required for estimating confidence band:** (G1-1, Table B2) TopP&R uses 10k real and fake samples, identical to all other metrics, and does not require a larger sample size to demonstrate its performance. In addition, TopP&R is efficient to compute (Table B2).
> &nbsp;
> **R3-W3. Critique existing metrics:** We have revised the manuscript to better emphasize the aspects suggested by the reviewer in our experimental results.
> &nbsp;
> **R3-W5. Experimental validation (showing TopP&R’s accuracy and consistency):** (G1, G2-1).

---

### Official Review · Reviewer_YKk3 · 2023-07-12

**Soundness:** 4 excellent
**Presentation:** 4 excellent
**Contribution:** 4 excellent
**Rating:** 8
**Confidence:** 4

**Summary:**

The paper discusses robust estimation of precision and recall in high dimensional and potentially noisy data. The approach relies on using kernel density estimation to approximate the underlying distributions and using a bootstrap estimation of confidence interval to ignore support with low probability mass. The authors show the effectiveness of this approach extensively using simulated and real data.

**Strengths:**

The paper is very well written and discusses the motivation and approach clearly. The problem addressed in the paper is very relevant as robust estimation of performance metrics is crucial for better comparing different approaches. The paper presents several interesting simulated and real examples to elaborate the method.

**Weaknesses:**

Although the estimation of confidence interval to remove 'noisy' samples is interesting, it would be great to see a discussion on estimating this uncertainty at individual points separately than working with the worst uncertainty, i.e., the inf norm over the kernel density.

**Questions:**

- Figure 1b: In the persistence plot, what does the shaded area imply? Can more blue points be added from the example on the right showing where in the persistence plot they land? Should the birth in this case always be at zero?
- Line 128: while c_X and c_Y are referred to as bootstrap bandwidth, is there a link between them and h_n and h_m, i.e., bandwidth of the kernel density estimate?
- Line 261: what does "swapping the position between Xi and Yj" imply?
- Line 275: sensitiveness or sensitivity?

**Limitations:**

Limitation of the method has not been discussed in details. More discussion on the computational complexity will be helpful.

---

> ### Author Rebuttal · Authors · 2023-08-09
>
> ## Reviewer 2
> **R2-1. Shaded area in Figure 1b:** For the sake of clarity in our explanation, it would be beneficial to provide a concise description of our Persistence Diagram (PD). At first glance, our PD might appear to be a swapping of the x-label and y-label, a confusion commonly associated with the PD derived from the kernel density estimator (KDE). **This confusion stems from the fact that the PD for KDE is generated through a 'super-level' filtration, distinct from the more typical 'sub-level' filtration used for most PDs, such as a Vietoris-Rips PD**. This distinction arises because KDE assigns high values to data points, making a 'decreasing' filtration value more appropriate. As a result, in this context, homological features are born at high values and die at low values, leading to a birth > death relationship. In particular, the PD for KDE almost never has homological features with birth = 0, while the PD for Vietoris-Rips has a lot of 0-dimensional features with birth = 0. To ensure comparability with the visualization of the PD from sub-level filtration, it has become customary to position death on the x-axis and birth on the y-axis.
>
> The shaded area corresponds to the orange area and the light green area in Figure A1. The orange area is where homological features have their birth < $c_\alpha$ and death < $c_\alpha$, and they usually correspond to outliers. The light green area is where homological features have their birth > $c_\alpha$ and death > $c_\alpha$, and they usually correspond to homological insignificant features lying within the estimated support. For either case, the shaded area corresponds to homological features that are statistically insignificant, and can be either (1) from noise, or (2) from signal but not eminent enough compared to the sample size. See Section C for a more detailed discussion.
> &nbsp;
> **R2-2. link between $c$ and $h$:** As in Algorithm 1 in **Section H.2**, the kernel bandwidth $h$ performs a function akin to the parameter $k$ employed in P&R and D&C. Notably, the parameter $k$ can influence the inherent breadth of the support, thereby affecting its scope. Employing a KDE that incorporates this parameter, the probability values of data features are measured. Subsequently, during the support estimation process, the probability values of features are used to approximate the bootstrap bandwidth ($c_X$ or $c_Y$). Thus, there exists a link between the kernel bandwidth $h$ and the bootstrap bandwidth, as they are intertwined in the approximation process.
> &nbsp;
> **R2-3. "swapping the position between $x_i$ and $y_j$":** This means the exchange of randomly selected datapoints $x_i\in\mathcal{X}$ and $y_j\in\mathcal{Y}$, such that each is transferred to the dataset to which other belongs (i.e., $x_i\in\mathcal{Y}$ and $y_j\in\mathcal{X}$).
> &nbsp;
> **R2-4. Sensitiveness or sensitivity?:** Thank you for your comment and we have revised that part.
> &nbsp;
> **R2-L1. Computation cost:** (Table B2) We have relatively similar computational complexity. We analyze the computational complexity and time consumption in Table B2 in the pdf we provide at this rebuttal.
> &nbsp;
> **R2-W1. Discussion on estimating this uncertainty at individual points separately:** Great point. As the reviewer pointed out, localizing the uncertainty at individual data points separately is an interesting direction of work, and it will be more sample efficient and disregard fewer data points. However, considering that we are preserving the topological signals, such a direction of localizing the uncertainty is almost hardly possible given the state of the art. For example, it is possible to localize the uncertainty of KDE $\hat{p}$, since the functional variability is local; to control the uncertainty of $\hat{p}$ at some point $x$, (roughly saying) we just need to analyze the function value at that point $\hat{p}(x)$. However, to localize the uncertainty of the homological feature lying at some point $x$, we need to analyze all the points that the homological feature lies on; hence even if we only want to control the uncertainty at a local point, it necessitates controlling the uncertainty on the global structure. This makes localizing the uncertainty for topological features difficult given the tools in topology and statistics we currently have. We have incorporated this discussion into the limitations section.

---

### Official Review · Reviewer_PyMg · 2023-07-26

**Soundness:** 2 fair
**Presentation:** 3 good
**Contribution:** 2 fair
**Rating:** 4
**Confidence:** 4

**Summary:**

The paper addresses the need for more reliable evaluation metrics for deep generative models. The existence of noise in real data, such as mislabeled data and adversarial examples, affects the reliability of existing metrics and may lead to false impressions of improvement when developing generative models. To overcome these issues, the paper proposes a new evaluation metric based on the idea of Topological Data Analysis (TDA) and statistical inference. The proposed metric uses persistent homology from TDA and bootstrap bandwidth with confidence level to distinguish between significant patterns (signal) and noise in the data, providing a more robust and reliable measure of generative model performance.

**Strengths:**

Given the recent substantial advancements in generative models, having a metric that is robust and reliable under various real-world noises holds significant importance. The authors present an approach that aims to overcome the limitations of previous metrics, using straightforward yet well-founded methods. Additionally, the paper introduces theoretical properties that show the metric's consistency in the presence of adversarial noises. The paper is well-written and easy to follow.

**Weaknesses:**

1. If I understand correctly, the authors use random linear projection only for the proposed metric, which might be unfair since it is unclear whether the improvements in robustness to outliers or noises stem from the low-dimensional space or the proposed methods. A comparison with previous methods operating in the same low-dimensional space would be more comprehensive.

2. The proposed metric's algorithm relies on multiple iterations for KDE filtration, raising questions about its practicality. It would be beneficial to include a comparison of the computational cost and overall runtime between the proposed metric and previous ones. Gaining insights into its efficiency compared to existing metrics will be valuable for both potential users and researchers. Additionally, it is essential to consider whether the metric can efficiently scale to handle a large number of datasets. However, the paper does not seem to provide details about the number of datasets used in the experiments, which leaves uncertainty about its scalability.

3. In section 5.1.2, the authors assert that precision values smaller than 1 indicate a problem. However, this lacks sufficient justification especially with a small number of samples. Additionally, the behavior of TopP for simultaneous mode dropping in Figure 3 exhibits worse instability compared to other metrics.

4. In the experimental results for mode dropping experiments with CIFAR-10, TopR does not exhibit consistent behavior with the toy experiments in the main paper (Figure 4) and sometimes performs worse than Recall and Coverage.

5. In section 5.2.3, the authors argue that the proposed metric is more sensitive to different levels of diverse noise, similar to FID. However, the results in Figure 6 do not appear to provide compelling evidence for this claim, except for the Gaussian noise case. Notably, in the salt and pepper noise case, TopPR, PR, and DC exhibit equal behavior, and in the black rectangle scenario, TopPR performs worse than PR. To strengthen the authors' claim about the proposed metric's sensitivity to diverse noise levels, additional explanation and experiments are necessary.

6. The sole reliance on FID as the ground truth ranking for models in section 5.2.4 might be insufficient and unconvincing. It is not clear how much relying on FID as GT is reliable. Including more widely used metrics, such as KID or human evaluations, would enhance the credibility of the result. However, even with the FID, the results do not seem to show the clear superiority of the proposed metric. Additionally, the results with HD across different embeddings also raise questions about the effect of the random projection layer, as other metrics do not utilize such a layer.

7. Some statements in the paper are unfounded. For instance, the rationale behind the ideal metric returning zero (line 263) is not clear. Moreover, the assumption in line 278 ("ground truth diversity should linearly decrease") is not correct because the supp(Q) remains the same with simultaneous mode dropping. In that sense, TopR works the worst of all.


**Questions:**

1. This paper employs the median of pairwise k-nearest distances between samples to estimate fixed-width kernels for KDE. I am curious whether this method is as effective as using sample-adaptive varying kernels, especially for datasets with multi-modal or long-tailed distributions. Adaptive kernels that adjust to local data characteristics might offer advantages in capturing complex and diverse patterns.

2. Concerning the experiments presented in Table 1, which dataset was used for these experiments? Can the proposed metric demonstrate effectiveness when applied to large-scale datasets, such as ImageNet?

3. There are some recommended experiments that are missing in the paper but would strengthen the effectiveness of the proposed metric:
- Evaluating metrics on Diffusion models such as ADM or EDM, which are the leading and prominent advancements in generative models, would be beneficial to demonstrate the practicality of the proposed metrics.
- Experiment for being robust to outliers in real-world dataset as in D&C paper.
- Ablation study regarding the hyperparameters alpha and h for real-world datasets.

4. To enhance the readability of the paper, the captions of figures should be more detailed.


**Limitations:**

Limitations are discussed in the supplement. Potential negative social impact is unlikely for a paper comparing distributions.

---

> ### Author Rebuttal · Authors · 2023-08-07
>
> ## Reviewer 1
> **R1-1. Adaptive kernels.** (G4)
> &nbsp;
> **R1-2, W2.**
> **The dataset utilized in Table 1:**  (G1-2) CIFAR-10. In the revised version, we have mentioned this in the caption.
> **Effectiveness on a large-scale dataset:**  (G1) Yes, it scales well. For all the experiments, we used 10k real and 10k fake samples, a standard choice for most of the evaluation metrics. We have added this detail to our paper.
> **Computational complexity:** (Table B2) TopP&R takes approximately 1 min 58 seconds of wall clock time for a single measurement on 10k samples, showing an efficient computational speed for model comparison. We have added this to our supplementary material.
> &nbsp;
> **R1-3.**
> **The outlier experiment similar to D&C:** (Section 5.2.2 and Figure 5) Please note that it has already been conducted in a completely identical manner, using outliers and inliers from the FFHQ dataset.
> **Comparison of state-of-the-art generative model** (G1-3)
> **Ablation study concerning hyper-parameters:** (G3)
> &nbsp;
> **R1-4. Improve captions:** Thank you for bringing this to our attention. In our revised paper, we have now updated the captions.
> &nbsp;
> **R1-W1, W6. Add KID:** (G1-2, G1-3), **Impact of random projection:** (G2)
> &nbsp;
> **R1-W3. Justification to precision values smaller than 1 indicate a problem:** Consistent with our results, Naeem et al. (2020) noted that precision does not approach 1, even when comparing two sets of samples from the same distribution, **irrespective of the sample size**. This makes determining a meaningful precision value difficult when assessing actual generative models. While TopP may exhibit cases where it does not converge to 1, it still shows a consistent evaluation trend, and it is difficult to argue that TopP values less than 1 significantly differ from the values that should converge. The key aspect of this experiment is the sensitivity of TopR to small changes in the mode. As the reviewer pointed out, even if TopP shows minor variations in this experiment, we believe that in many of our other experiments, TopP&R demonstrated more advantages than drawbacks.
> &nbsp;
> **R1-W4. Mode dropping experiments on CIFAR-10 are not strong:** (G1-4) We believe that the reviewer is referring to Figure 3 (not Figure 4). We agree with the reviewer that, compared to the results of Figure 3, it is hard to say that TopR is clearly better than the others in CIFAR-10 experiments. To further investigate the differences between TopP&R and the existing metrics, we conducted experiments using a large-scale dataset, baby imagenet, which has higher image resolution and diversity than CIFAR-10. In this large-scale dataset, TopP&R and the other metrics exhibit similar trends as seen in the toy data experiment–TopP&R shows sensitivity to distribution changes and exhibits precise responses.
> &nbsp;
> **R1-W5. The authors argue that the proposed metric is more sensitive to different levels of diverse noise, similar to FID:** We wish to clarify that we did **not** assert that TopP&R is "more" sensitive to varying levels of diverse noise compared to all other metrics. Our comparison was limited to MTD, a topology-based method like ours, but one that exhibited inconsistent reactions to the distortions. Other than that, as it can be seen in Section 5.2.3, we simply claimed that TopP&R effectively captures the various levels of distortion introduced to the images. Our objective was to demonstrate that TopP&R offers a consistent evaluation across different noise levels. We did not intend to suggest that our metric shows the best response.
> &nbsp;
> **R1-W7.
> The rationale behind the ideal metric returning zero (line 263):** As the reviewer pointed out, it is indeed appropriate to modify the statement in line 263 to “has low values” (not zero).
>
> **Assumption in line 278  ("ground truth diversity should linearly decrease") is not correct:**
> It is correct. The $supp(Q)$ shrinks as the number of samples of each mode simultaneously decreases. This is because the samples are from the Gaussian distribution of high dimensional space, where they are sparsely distributed. Hence the deletion of the samples actually results in shrinking the support, which leads to a linear decrease in ground truth diversity.
> This experiment aligns with Naeem et al.’s (2020) work, aiming to achieve an ideal linear response to simultaneous mode drop. The experiment in Section 5.2.1 starts with identical $supp(P)$ and $supp(Q)$ and progressively reduces samples of nine modes within $supp(Q)$ in a linear fashion until these nine modes vanish, while replenishing the first mode with the decreased number of samples. We would like to clarify that there might have been a misunderstanding regarding this aspect. Although the total sample count within $supp(Q)$ remains unchanged, the distribution itself experiences the linear disappearance of nine modes, ultimately leaving only one mode. Thus, in terms of diversity, the reduction indeed follows a linear trend.  Therefore, it is crucial for the metric to capture the linear reduction in diversity within $supp(Q)$. The results in G1-4 of general comments further confirm that TopP&R responds most effectively to such linear trends.

---

> > ### Comment · Reviewer_PyMg · 2023-08-13
> >
> > R1-W7: Do you mean that supp(Q) changes depending on the number of samples? Is it in accordance with the definition of supp() in line 74? I believe the support of a probability distribution Q should depend on Q itself, not the observations. In the experiment, we precisely know the probability distributions, and their supports are not changed.

---

> > > ### Author Response · Authors · 2023-08-14
> > >
> > > We are sorry for the confusion, and as the reviewer pointed out, the ground truth support $supp(Q)$ remains unchanged regardless of the samples, provided that we use the definition supp(Q) as it is. However, what we meant by "the $supp(Q)$ shrinks" indeed contains subtle implications that were not possible to be precisely explained in the previous rebuttal due to the word limit. So we will explain here:
> > >
> > > When we think of what should be the "ideal ground truth diversity", for most of the scenarios it should be the population recall, i.e., $recall_{P}(Q)=P(supp(Q))$. This is because we usually think of the ideal scenario as the asymptotic case, i.e., $n, m \to \infty$, and then the data approximately give the full information as the distribution. For this case, we have already shown that TopP and TopR are consistent with precision and recall, respectively, in Proposition 4.1 and Theorem 4.2.
> > >
> > > However, the scenario in 5.2.1 is a little different: for this case, technically speaking, $supp(P)=supp(Q)=\mathbb{R}^{d}$, so diversity would be always $1$, but this would not be the desired result. In detail, although the total sample size is the same, the sample sizes of nine modes are simultaneously decreasing. So for the data belonging to these 9 modes, it is equivalent to saying that $m \to 0$. When we have a small sample, and when the support is high dimensional, it is impossible to recover the support from the sample. This is because filling the high dimensional support with the data is impossible. Hence, not only TopP&R but also with any estimator of support, it is not possible to recover the full support $supp(Q)$. With the sample size $m$, what we can recover at most is the partial support $\{q>C_{m}\}$, where $q$ is the density of $Q$ and $C_{m}$ is a sequence that decreases as $m$ increases. Hence, as the sample sizes of nine modes are decreasing, "the partial support" $\{q>C_{m}\}$, in the sense of the ideal support that we can recover with the given number of data, shrinks. This is what we precisely meant by "the $supp(Q)$ shrinks". And then the "ideal ground truth diversity", in the sense that any good estimator of the diversity should approach, also correspondingly decreases.
> > >
> > > This “ideal ground truth diversity” also coincides with the recent trend in generative models. As mentioned by Han et al. (2022), recent generative models aim to adequately generate a sufficient number of diverse samples within each mode of the real data distribution. In this regard, an evaluation model that fails to accurately capture the reduced sample size of mode does not align with current trends in generative model research. Developing a metric that accurately captures this tendency is very important. The aforementioned “ideal ground truth diversity” aligns well with the recent research direction, and so is TopP&R. TopP&R becomes an indispensable tool for evaluating the ability of newly developed generative models to generate samples diversely and accurately under statistical confidence. We will clearly elucidate how the ideal ground truth diversity should behave and the connection to the recent research trend into our paper.
> > >
> > > &emsp; - Han, Jiyeon, et al. "Rarity score: A new metric to evaluate the uncommonness of synthesized images." arXiv preprint arXiv:2206.08549 (2022).

---

> > > > ### Comment · Reviewer_PyMg · 2023-08-14
> > > >
> > > > "diversity would be always 1."
> > > > => This is not true. When the concentration on the first mode is 1.0, diviersity theoretically becomes 1/10, which both Recall and GCA recall quite accurately approximate.
> > > > In other words, according to the definition of diversity in the paper, TopR performs the worst compared to previous works, as I pointed out in my review.
> > > > If you think the "ideal ground truth diversity” differs from your definition of recall, please consider changing the definition accordingly.

---

> > > > > ### Author Response · Authors · 2023-08-16
> > > > >
> > > > > We appreciate the reviewer for identifying ambiguities in our original manuscript and in our response. This feedback has significantly improved our understanding and the quality of the manuscript. Before we proceed, we would like to clarify a few points.
> > > > > &emsp;
> > > > > (1) Firstly, **when we stated '$supp(P)=supp(Q)=\mathbb{R}^{d}$, so the diversity would always be 1'**, we were referring to the mixture of Gaussian distributions. For this case, $supp(Q)$ would be $\mathbb{R}^d$ even in the event of a complete mode collapse. For this case, the "ideal diversity" $recall_{Q}(P)=P(supp(Q))$ is always 1 even at the complete mode collapse. On the other extreme is the case where each distribution of an individual mode has disjoint support; for this case, the "ideal diversity" $P(supp(Q))$ maintains 1 when the mode concentration $\in [0, 1)$, and at the complete mode collapse it suddenly drops to 1/(number of modes). Hence there is already a problem with the existing definition of "ideal diversity", which does not match well with the recent research trends that we described in the previous comments.
> > > > > Recent generative models aim to adequately generate a sufficient number of diverse samples within each mode of the real data distribution. With this perspective, the "ideal diversity" that maintains 1 when the mode concentration $\in [0, 1)$ becomes useless as an ideal evaluation metric. An evaluation metric should be able to accurately assess a generator's distribution that might not fully capture all modes of the real distribution at their actual ratios, and where some mode dropping occurs. While the value itself of each metric is essential, it is equally vital to gauge how sensitively the metric responds as the mode drop intensifies. In this context, the simultaneous mode drop experiment was conceived in the previous D&C paper [Naeem et al 2020], which we followed. **In this regard, we also agree that, as the reviewer rightly pointed out, the definition of diversity must be changed accordingly**. We will clarify the definition of Diversity to align it better with the original intent of the experiment, minimizing potential misunderstandings.
> > > > > &emsp; - Naeem, Muhammad Ferjad, et al. "Reliable fidelity and diversity metrics for generative models." International Conference on Machine Learning. PMLR, 2020.
> > > > > &emsp;
> > > > > (2) **From the reviewer's discussion on P&R and GCA recall**, it seems that the reviewer is referencing the results from our toy dataset presented in Figure 3, Section 5.1.2. If that is the case, we would like to clarify that we used the Mixture of Gaussians with seven modes, not 10. Upon a complete collapse of all modes into the primary mode, the metrics showed values as: TopR at 0.1065, recall at 0.0068, coverage at 0.1411, and GCA recall at 0.1351. Assessing solely from the values during this final collapse phase, the Recall metric appears to underperform compared to GCA recall and coverage. Yet, in line with our goal of diversity (as outlined in point (1)), **the core objective of this experiment is to ascertain the capability of the diversity metric to consistently detect the progressive loss of modes**. Additionally, beyond this simultaneous mode drop experiment, we would like to also highlight that our metric has shown consistent and strong performance in various other contexts.

---

### Author Rebuttal · Authors · 2023-08-07

We deeply appreciate your feedback. Your insights greatly helped us to refine our presentation and better illuminate the key advantages of our metric. Your input has also greatly enhanced our discussion about the metric's limitations, which not only clarifies our current work but also highlights promising future directions.

**Reviewers can find the additional experiments we conducted in the attached PDF file: Table B1\~5 and Figure B1\~5.**
&nbsp;
**G1. Detailed info. of Table 1 & Scalability to large-scale data with more recent generative models (ADM, StyleGAN-XL)**

**G1-1.** All the evaluation in this paper was conducted using **10k real and 10k fake samples**.

**G1-2. Model ranking experiment (Table 1):** The models were trained on **CIFAR-10**. Here, our aim was to highlight the consistency of TopP&R across different embeddings and its congruence with widely accepted metrics in the field, FID. Based on the reviewer's recommendation, we additionally included **KID (Table B1)**. A foundational aspect of an evaluation metric is its capacity for stability and consistency. Ideally, the ranking it provides for different models should remain largely stable, irrespective of the embedding network used. Our results show that TopP&R consistently outperforms in terms of stability, as evidenced by the MHD values across different embedding networks (TopP&R: 1.33, P&R: 2.66, D&C: 3.0, MTD: 3.33). Furthermore, as generative models have developed with an emphasis on aligning with FID's (or KID’s), we posit that the rankings determined by FID (or KID) can, to some extent, reflect the true performance hierarchy among models, even if it doesn't capture it perfectly. Our results show that TopP&R displays the strongest alignment with FID and KID, which resonates with the historical trajectory of model development.

**G1-3. Ranking SOTA models (Table B3):** We evaluated generative models trained on **ImageNet**, including **ADM and StyleGAN-XL**. The results consistently show that TopP&R maintains stable rankings across various embeddings and aligns most closely with FID and **KID**.

**G1-4. Mode dropping on Baby ImageNet (Figure B1):** In contrast to the CIFAR-10 outcomes where TopP&R produced results comparable to other metrics, **with the large-scale dataset**, both the sequential and simultaneous mode dropping scenarios highlight **TopP&R's precise assessment, which is in line with our observations from the toy experiments**. Conversely, P&R had difficulties in accurately evaluating the simultaneous mode drop.
&nbsp;
**G2. Random projection (RandProj)**
**G2-1. The impact of RandProj (Table B4, Figure B3 and B4):** To discern if TopP&R's enhanced performance stems from RandProj, we applied RandProj to both P&R and D&C. This led to inconsistent results in model ranking (Table B4), suggesting that **the performance of TopP&R is attributed to its distinct features, rather than just the RandProj**. This trend was further observed in the "shifting the generated feature manifold (Section 5.1.1)" (Figure B3), and the "Tolerance to Non-IID perturbations (Section 5.1.3)" (Figure B4). Both experiments were conducted with and without RandProj. Across these studies, a consistent theme emerged: both P&R and D&C failed to exhibit a stable evaluation trend in relation to noise, irrespective of the use of RandProj.

**G2-2. The effect of dimensionality (Figure B5):**
We examine the **impact of changing the dimension of RandProj** on TopP&R. We employed the same experimental setup as the "Mode dropping experiment (Figure B1)" and observed how the evaluation trend of TopP&R changes by altering the dimension of RandProj. By Johnson-Lindenstrauss Lemma (Johnson et al. 1986), it is known that RandProj preserves the topological structure of data, given the output dimension isn't excessively reduced (see Section G.3). When we use higher dimensions than 32, our results show that the trend of TopP&R stays consistent and stable. However, using higher dimensions leads to greater computational overhead. Thus, for pragmatic computation purposes, we opted for a dimension of 32.
&nbsp;
**G3. Choosing parameters of TopP&R**
**G3-1. Confidence level of bootstrap bandwidth ($\alpha$):** $\alpha$ is not a typical tuning hyperparameter, but it holds a statistical interpretation of the probability or degree of confidence in accommodating errors, noise, and similar factors. The most commonly chosen values include $\alpha=0.1, 0.05, 0.01$, corresponding to confidence levels of 90%, 95%, and 99% respectively. In our experiments, we consistently employed $\alpha = 0.1$.

**G3-2. Kernel bandwidth ($h$):** $h$ serves to adjust the initial KDE probability values of features. $h$ being too small leads to the KDE with unstable topological features, and $h$ being too large leads to the KDE with its topological features dissolved, so $h$ should be properly chosen. Ideally, $h$ can be chosen using topologically significant features, but this incurs computational challenges. Instead, in Algorithm 2 of Section H.4 we use a heuristic and automatic approximation of $h$ that effectively adapts to various embeddings and arbitrary datasets. From various experimental results, this $h$ performs effectively in practice.
&nbsp;
**G4. Different methods rather than KDE**
Please see our Remark 4.4. The super-level set of KDE excludes topological noise by the KDE filtration: a confidence band establishes a statistical threshold for distinguishing between signal and noise in the KDE filtration. However, if we switch from using the super-level set of KDE to other methods like k-NN or any alternative, the estimated support is not a result of KDE filtration. Consequently, the topological noise present cannot be assessed through KDE filtration, and the confidence band cannot be applied to determine signal or noise in support. Consequently, the guarantee of excluding topological noise is lost when using different methods; this is not known yet.

---

### Decision · Program_Chairs · 2023-09-21

**Decision:**

Accept (poster)

**Comment:**

This paper develops a novel, promising approach to evaluating deep generative models. The authors have done a good job addressing all of the reviewer's concerns.